# Influential Bandits:
# Pulling an Arm May Change the Environment

**Ryoma Sato**                                                                 *rsato@nii.ac.jp*
*National Institute of Informatics*

**Shinji Ito**                                                      *shinji@mist.i.u-tokyo.ac.jp*
*The University of Tokyo and RIKEN*

**Reviewed on OpenReview:** *https://openreview.net/forum?id=YNKaDfYbY3*

## Abstract

While classical formulations of multi-armed bandit problems assume that each arm's reward is independent and stationary, real-world applications often involve non-stationary environments and interdependencies between arms. In particular, selecting one arm may influence the future rewards of other arms, a scenario not adequately captured by existing models such as rotting bandits or restless bandits. To address this limitation, we propose the *influential bandit* problem, which models inter-arm interactions through an unknown, symmetric, positive semi-definite *interaction matrix* that governs the dynamics of arm losses. We formally define this problem and establish two regret lower bounds, including a superlinear $\Omega(T^2/\log^2 T)$ bound for the standard LCB algorithm (loss minimization version of UCB) and an algorithm-independent $\Omega(T)$ bound, which highlight the inherent difficulty of the setting. We then introduce a new algorithm based on a lower confidence bound (LCB) estimator tailored to the structure of the loss dynamics. Under mild assumptions, our algorithm achieves a regret of $O(KT \log T)$, which is nearly optimal in terms of its dependence on the time horizon. The algorithm is simple to implement and computationally efficient. Empirical evaluations on both synthetic and real-world datasets demonstrate the presence of inter-arm influence and confirm the superior performance of our method compared to conventional bandit algorithms.

## 1 Introduction

The multi-armed bandit problem [7, 20] and its extensions serve as fundamental models for decision-making under uncertainty. In such problems, one aims to maximize the cumulative reward while simultaneously learning the underlying reward model through repeated selection of actions referred to as arms and observation of the resulting rewards in an environment where the reward distributions are initially unknown. In the most basic setting, it is assumed that the reward for each arm is independently drawn from an unknown, time-invariant (i.e., stationary) distribution. A variety of algorithms, including Upper Confidence Bound (UCB) [2, 19] and Thompson Sampling [1, 26], have been proposed for this setting, and their effectiveness has been validated both theoretically and empirically.

In real-world applications, however, rewards are not necessarily stationary and may be influenced by past actions. For example, in recommender systems, arms correspond to items being recommended, and rewards represent user feedback or engagement with those items. Recommending the same item repeatedly may lead to user fatigue, resulting in decreased engagement. Conversely, repeated recommendations could increase a user's interest in the item, thereby enhancing their response over time. To model such dynamics, various models and algorithms have been proposed, including restless bandits [25, 29], rotting bandits [21, 24] and rising bandits [14]. In these models, the state transitions for each arm are modeled to depend on the arm selection, and the reward distribution is assumed to vary depending on the state. Other approaches have

also been explored for instance, the blocking bandit framework [5, 6] introduces constraints that prohibit consecutive selections of the same arm, which can be interpreted as a way to address reward decay.

One of the key limitations of these existing models lies in the strong assumption that the impact of selecting an arm is confined solely to the reward of that particular arm. Due to this limitation, these models fail to capture scenarios in which selecting one arm may influence the future rewards of other arms. In real-world problems, it is rare for the rewards of all arms to be completely independent, making it important to consider such interdependencies. For instance, in the context of recommender systems, recommending a particular item (such as a movie or product) and having the user engage with it may reduce their interest in similar items, or conversely, increase their interest in related items.

To overcome this limitation and address scenarios where interactions between arms exist, as described above, we initiate the study of the *influential bandit* problem. We begin by formally defining the influential bandit problem and deriving a lower bound on the regret, thereby highlighting the inherent difficulty of the problem. We then propose an algorithm tailored to this setting and provide a theoretical analysis of its performance. Finally, we validate the proposed model and assess the effectiveness of the algorithm through empirical experiments using synthetic and real-world data.

## 1.1 Our contribution

In formulating the influential bandit problem, we introduce an *interaction matrix* $\boldsymbol{A} \in \mathbb{R}^{K \times K}$ to model the influence that selecting one arm has on the rewards of other arms, where $K$ denotes the number of arms. This matrix captures the pairwise interactions between arms, where each element represents the impact of selecting arm $i$ on the loss of arm $j$. We assume that the interaction matrix is symmetric and positive semi-definite. The matrix is initially unknown, and addressing the uncertainty in the environment arising from this is a critical challenge in this problem. For performance evaluation, we define regret as the difference between the cumulative loss incurred by the algorithm and the cumulative loss of the optimal sequence of actions, assuming full knowledge of both the interaction matrix and the initial loss values.

To illustrate the difficulty of this problem, we present two regret lower bounds. First, we show that the LCB algorithm (loss minimization version of UCB [2]), one of the most standard algorithms for the classical stochastic bandit problem, suffers a regret of $\Omega(T^2/\log^2 T)$ in the worst case, where $T$ denotes the time horizon. At first glance, a superlinear regret in $T$ may appear surprising in the context of traditional bandit problems; however, such behavior is common under the setting considered in this work. Indeed, since each arm selection can influence future losses, the per-round loss can grow as large as $O(T)$, leading to a maximum possible regret of $O(T^2)$. Therefore, achieving regret strictly smaller than $O(T^2)$ is considered a non-trivial result. As a second lower bound, we show that any algorithm must incur at least $\Omega(T)$ regret in the worst case. This implies that, from the perspective of worst-case analysis (ignoring constant factors), achieving $O(T)$ regret is optimal.

One of the main contributions of this paper is the proposal and validation of a novel algorithm for the influential bandit problem. The proposed method is based on a new lower confidence bound (LCB) estimator that accounts for the structure of the loss dynamics. Under a mild assumption that the noise is bounded, we show that the algorithm achieves a regret of $O(KT \log T)$, where $K$ is the number of arms and $T$ is the time horizon. In comparison with the previously established lower bounds, this result demonstrates that the proposed method outperforms the standard LCB algorithm and is nearly optimal in terms of its dependence on $T$. Furthermore, the algorithm is easy to implement and computationally efficient, making it practically useful as well.

The effectiveness of the proposed model and algorithm is also evaluated through numerical experiments. First, experiments on synthetic data confirm that the proposed algorithm consistently outperforms the standard LCB algorithm in practice. Next, using a real-world dataset, we verify the presence of inter-arm interactions as modeled in our framework and investigate the specific nature of these interactions. We further demonstrate the superiority of the proposed algorithm through empirical evaluation.

## 2 Influential Bandits

Let $\boldsymbol{A} \in \mathbb{R}^{K \times K}$ be an unobserved interaction matrix, which is positive semi-definite, and $l_j^{(1)} \in \mathbb{R}$ be the unobserved initial loss for each arm $j \in [K]$. We consider the following online learning problem. An algorithm repeatedly chooses an arm $i^{(t)} \in [K]$ at time $t$, and observes a loss $l_{i^{(t)}}^{(t)} + \xi^{(t)}$, where $\xi^{(t)}$ is a random noise with zero mean and bounded variance. The losses are then updated by

$$l_j^{(t+1)} = l_j^{(t)} + \boldsymbol{A}_{i^{(t)}j} \quad \text{for all } j \in [K]. \tag{1}$$

The goal is to minimize the regret, which is defined as the difference between the total loss of the algorithm and the total loss of the best sequence of actions in hindsight. Note that the benchmark is the best sequence of actions in hindsight, not the best single arm in hindsight. We call this problem the influential bandit problem. The crux of this problem is the interaction matrix $\boldsymbol{A}$, which typically represents the similarity between arms. When we pull an arm, the expected loss of similar arms increases. For example, in the online advertising problem, repeating the same or similar advertisements may decrease the effectiveness of the advertisements and thus increase the loss. Some elements can be negative, which models the recovery effect. For example, repeating horror movies may lead to fatigue, but sandwiching a horror movie between two comedy movies may recover the effectiveness of the horror movie, i.e., $A_{\text{horror,horror}} > 0$ but $A_{\text{comedy,horror}} < 0$. The influential bandit problem is a generalization of the stochastic bandit problem, where the interaction matrix is the null matrix.

We first show the following fundamental property of the influential bandit problem.

**Proposition 2.1.** *Let $\boldsymbol{x} = \sum_{t=1}^{T} \boldsymbol{e}^{(i^{(t)})}$ be the total number of times each arm is chosen, where $\boldsymbol{e}^{(i)} \in \mathbb{R}^K$ is the one-hot vector of arm $i$. The expected total loss is given by*

$$\sum_{t=1}^{T} l_{i^{(t)}}^{(t)} = \boldsymbol{l}^{(1)\top} \boldsymbol{x} + \frac{1}{2} \boldsymbol{x}^\top \boldsymbol{A} \boldsymbol{x} - \frac{1}{2}[\boldsymbol{A}_{11}, \boldsymbol{A}_{22}, \ldots, \boldsymbol{A}_{KK}] \boldsymbol{x}. \tag{2}$$

*In particular, the total loss does not depend on the order of the actions.*

*Proof.* The expected loss of arm $j$ at time $t$ is given by

$$l_j^{(t)} = l_j^{(1)} + \sum_{s=1}^{t-1} \boldsymbol{A}_{i^{(s)}j} \tag{3}$$

for all $j \in [K]$. The expected total loss is then given by

$$\sum_{t=1}^{T} l_{i^{(t)}}^{(t)} = \sum_{t=1}^{T} l_{i^{(t)}}^{(1)} + \sum_{t=1}^{T}\sum_{s=1}^{t-1} \boldsymbol{A}_{i^{(s)}i^{(t)}} \tag{4}$$

$$= \boldsymbol{l}^{(1)\top} \boldsymbol{x} + \sum_{t=1}^{T}\sum_{s=1}^{t-1} \boldsymbol{A}_{i^{(s)}i^{(t)}} \tag{5}$$

$$\overset{\text{(a)}}{=} \boldsymbol{l}^{(1)\top} \boldsymbol{x} + \frac{1}{2} \left( \sum_{s=1}^{T}\sum_{t=1}^{T} \boldsymbol{A}_{i^{(s)}i^{(t)}} - \sum_{t=1}^{T} \boldsymbol{A}_{i^{(t)}i^{(t)}} \right) \tag{6}$$

$$= \boldsymbol{l}^{(1)\top} \boldsymbol{x} + \frac{1}{2} \boldsymbol{x}^\top \boldsymbol{A} \boldsymbol{x} - \frac{1}{2}[\boldsymbol{A}_{11}, \boldsymbol{A}_{22}, \ldots, \boldsymbol{A}_{KK}] \boldsymbol{x}, \tag{7}$$

where (a) follows from the symmetry of $\boldsymbol{A}$. The last equation shows that the total loss does not depend on the order of the actions but only on the total number of times each arm is chosen. $\square$

Let $\boldsymbol{b} = \boldsymbol{l}^{(1)} - \frac{1}{2}[\boldsymbol{A}_{11}, \boldsymbol{A}_{22}, \ldots, \boldsymbol{A}_{KK}]^\top$. Then, the expected total loss is given by

$$\mathcal{L}(\boldsymbol{x}) = \boldsymbol{b}^\top \boldsymbol{x} + \frac{1}{2} \boldsymbol{x}^\top \boldsymbol{A} \boldsymbol{x}. \tag{8}$$

Therefore, the influential bandit problem is equivalent to finding the sequence of actions that minimizes the quadratic loss in Eq. 8, in contrast to the stochastic bandit problem, where the loss is linear. The challenges lie in the following facts.

- Neither $\boldsymbol{b}$ nor $\boldsymbol{A}$ is known in advance.

- Choices of actions are not independent. One action affects the future losses of other actions. The negative effect of an exploratory action may persist over time. We will present an example where one failure choice causes catastrophic consequences in Proposition 2.3.

- In particular, we cannot arbitrarily choose $\boldsymbol{x}$ but must sequentially update $\boldsymbol{x} \leftarrow \boldsymbol{x} + \boldsymbol{e}^{(i^{(t)})}$ and thus cannot drastically change the optimization variable $\boldsymbol{x}$ at each time step. This introduces a challenge from the optimization perspective.

One of the key distinctions between the influential bandit problem and the stochastic bandit problem lies in the loss scale. Since the influential bandit problem incurs quadratic loss, expecting sublinear regret is unrealistic but subquadratic regret is a typical objective, whereas in standard linear bandits, achieving sublinear regret is a typical objective. Indeed, even $O(T^{1.99})$ regret is non-trivial and the standard lower confidence bound (LCB) algorithm, which corresponds to the upper confidence bound (UCB) algorithm in the reward maximization setting, cannot achieve this.

**Proposition 2.2.** *There exists an instance of the influential bandit problem such that the standard LCB algorithm, which selects*

$$i^{(t)} = \arg\min_{i \in [K]} \left( \hat{\mu}_i - \sqrt{\frac{2 \log t}{n_i}} \right) \tag{9}$$

*and selects the minimum index when there are ties for all $t \in [T]$, incurs the regret $\Omega\left(\frac{T^2}{\log^2 T}\right)$, where $\hat{\mu}_i$ is the empirical mean of the losses of arm $i$ and $n_i$ is the number of times arm $i$ is chosen.*

We prove the proposition by construction. Notably, the counterexample instance is noiseless. The LCB algorithm fails even in a noiseless environment in the influential bandit problem.

*Proof Sketch.* The counter example is $K = 2, \boldsymbol{A} = [[1, 1], [1, 2]], \boldsymbol{l}^{(1)} = [1, 1]$, and $\xi^{(t)} = 0$. There are no noises, and the process is deterministic. The optimal sequence of actions is $i^{(t)} = 1$ for all $t \in [T]$, which incurs the total loss $\frac{1}{2}T(T + 1)$. We can prove that the number of times the LCB algorithm chooses arm 2 is at least $\Omega\left(\frac{T}{\log T}\right)$, which leads to $\Omega\left(\frac{T^2}{\log^2 T}\right)$ regret. The key intuition behind the lower bound is as follows. Whenever the LCB algorithm selects the optimal arm (arm 1), the losses for both arm 1 and arm 2 are increased by the environment. However, only the loss of arm 1 is actually observed, so the LCB algorithm updates the estimated loss only for arm 1, while the estimate for arm 2 remains unchanged. After pulling arm 1 consecutively for more than about $O(\log t)$ rounds, the LCB score of arm 1 becomes larger than that of arm 2. This guarantees that arm 2 will be selected at least on the order of $\Omega\left(\frac{T}{\log T}\right)$ by the LCB algorithm. The complete proof is in Appendix A. □

In addition, there exists a counterexample where the regret must be at least linear in the time horizon.

**Proposition 2.3.** *There exists an instance of the influential bandit problem such that for any algorithm, the regret is at least $\Omega(T)$.*

*Proof.* Let $K = 2$. Since initially an algorithm only knows the number of arms and no further instance-specific information, its first action cannot depend on the instance. Suppose the first action

is $i^{(1)} = 1$. Let

$$\boldsymbol{A} = \begin{pmatrix} 1 & \frac{1}{2} \\ \frac{1}{2} & \frac{1}{4} \end{pmatrix} \quad \text{and} \quad \boldsymbol{b} = \begin{pmatrix} 0 \\ 0 \end{pmatrix}. \tag{10}$$

The optimal sequence of actions is $i^{(t)} = 2$ for all $t \in [T]$, which incurs the total loss $\frac{1}{8}T^2$. By contrast, the best possible sequence of actions after $i^{(1)} = 1$ is $i^{(t)} = 2$ for all $t = \{2, 3, \ldots, T\}$. This incurs the total loss $\frac{1}{8}T^2 + \frac{1}{4}T + \frac{1}{8}$. Therefore, the regret is at least $\frac{1}{4}T + \frac{1}{8} = \Omega(T)$. □

The counterexample above indicates that the influence of an action lasts, and a single failure choice may cause catastrophic consequences. If an action increases the loss of other actions by a constant, the total regret increases linearly in the time horizon.

We will propose an algorithm that achieves near-optimal regret. Before presenting the algorithm, we introduce the following assumption for simplicity.

**Assumption 2.4.** The maximum absolute value of the interaction matrix $\boldsymbol{A}$ is bounded by 1.

**Assumption 2.5.** The noise $\xi^{(t)}$ is bounded by 1.

The proposed algorithm is as follows.

---

**Algorithm 1:** Influential LCB

---
1   $c_i^{(1)} \leftarrow 0$   for all $i \in [K]$          // time passed since the last observation
2   $\hat{l}_i^{(1)} \leftarrow -\infty$   for all $i \in [K]$          // last observed loss
3   **for** $t = 1, 2, \ldots, T$ **do**
4      $i^{(t)} \leftarrow \arg\min_{i \in [K]} \hat{l}_i - c_i$          // lower confidence bound
5      Observe loss $L^{(t)} = l_{i^{(t)}}^{(t)} + \xi^{(t)}$
6      $\hat{l}_i^{(t+1)} \leftarrow \begin{cases} L^{(t)} & \text{if } i = i^{(t)} \\ \hat{l}_i^{(t)} & \text{otherwise} \end{cases}$          // update the estimate by the observed loss
7      $c_i^{(t+1)} \leftarrow \begin{cases} 1 & \text{if } i = i^{(t)} \\ c_i^{(t)} + 1 & \text{otherwise} \end{cases}$          // update the confidence bound

---

The expected loss of an arm decreases by at most one at each time step by Assumption 2.4. If we observe the loss $\hat{l}_i$ of arm $i$ at time $(t - c_i)$, a reasonable lower bound of the loss of arm $i$ at time $t$ is $\hat{l}_i - c_i - 1$. The regret of Algorithm 1 is given by the following theorem.

**Theorem 2.6.** *The regret of Algorithm 1 is at most*

$$\left( \frac{5K+3}{2} + 2\|\boldsymbol{l}^{(1)}\|_\infty \right) T + \left( 2K + 2\|\boldsymbol{l}^{(1)}\|_\infty + 4 \right) T \log T = O(KT \log T) \tag{11}$$

A failure attempt to prove this would be using the relation $\hat{l}_{i^{(t)}}^{(t)} - c_{i^{(t)}} - 1 \leq \hat{l}_{i^{(t)*}}^{(t)} - c_{i^{(t)*}} - 1 \leq l_{i^{(t)*}}^{(t)}$, where $i^{(t)*}$ is the optimal action at time $t$, summing over $t$, using the relation $\sum_t c_{i^{(t)}} \leq (K-1)T$, and concluding that the regret $\sum_t l_{i^{(t)}}^{(t)} - l_{i^{(t)*}}^{(t)}$ is at most $(K-1)T + T = O(KT)$. This is not valid because $l_{i^{(t)*}}^{(t)}$ is the loss in the very state the algorithm has reached with $i^{(1)}, i^{(2)}, \ldots, i^{(t-1)}$, but there might be a better sequence $i^{(1)*}, i^{(2)*}, \ldots, i^{(t-1)*}$ that leads a smaller loss at time $t$. The regret is not $\sum_t l_{i^{(t)}}^{(t)} - l_{i^{(t)*}}^{(t)}$ since the choices are not independent. We need to avoid independently treating states.

*Proof.* Let $i^{(1)}, i^{(2)}, \ldots, i^{(T)}$ be the sequence of actions chosen by Algorithm 1, $\boldsymbol{x}^{(t)}$ be the total number of times each arm is chosen up to time $t$, and $\mathcal{L}(\boldsymbol{x}^{(t)})$ be the total loss at time $t$. As the

combinatorial problem is cumbersome, we use the continuous solution

$$\boldsymbol{x}^{(t)*} = \arg\min_{\boldsymbol{x} \in t\Delta_K} \mathcal{L}(\boldsymbol{x}) \tag{12}$$

as the benchmark, where $t\Delta_K = \{\boldsymbol{x} \in \mathbb{R}^K \mid \boldsymbol{x} \geq 0, \boldsymbol{x}^\top \mathbf{1} = t\}$ is the $(K-1)$-dimensional probability simplex expanded by $t$. $\mathcal{L}(\boldsymbol{x}^{(t)*})$ is a lower bound of the total loss $\min_{\boldsymbol{x} \in t\Delta_K \cap \mathbb{Z}_*^K} \mathcal{L}(\boldsymbol{x})$ of the optimal sequence of actions. Let

$$\boldsymbol{p}^* = \arg\min_{\boldsymbol{p} \in \Delta_K} \frac{1}{2}\boldsymbol{p}^\top \boldsymbol{A}\boldsymbol{p} \tag{13}$$

be the optimal distribution of arms that minimizes the quadratic term $\mathcal{L}_2(\boldsymbol{p}) \stackrel{\text{def}}{=} \frac{1}{2}\boldsymbol{p}^\top \boldsymbol{A}\boldsymbol{p}$. Let $\boldsymbol{p}^{(t)} = \frac{\boldsymbol{x}^{(t)}}{t}$ be the empirical distribution of arms. We have

$$\boldsymbol{p}^{(t+1)} - \boldsymbol{p}^{(t)} = \frac{\boldsymbol{x}^{(t+1)}}{t+1} - \boldsymbol{p}^{(t)} \tag{14}$$

$$= \frac{\boldsymbol{x}^{(t)} + \boldsymbol{e}^{(i^{(t)})}}{t+1} - \boldsymbol{p}^{(t)} \tag{15}$$

$$= \frac{t\boldsymbol{p}^{(t)} + \boldsymbol{e}^{(i^{(t)})}}{t+1} - \boldsymbol{p}^{(t)} \tag{16}$$

$$= \frac{1}{t+1}\left(\boldsymbol{e}^{(i^{(t)})} - \boldsymbol{p}^{(t)}\right). \tag{17}$$

We then have, by Taylor expansion around $\boldsymbol{p}^{(t)}$,

$$\mathcal{L}_2(\boldsymbol{p}^{(t+1)}) = \mathcal{L}_2(\boldsymbol{p}^{(t)}) + \nabla\mathcal{L}_2(\boldsymbol{p}^{(t)})^\top\left(\boldsymbol{p}^{(t+1)} - \boldsymbol{p}^{(t)}\right) + \frac{1}{2}\left(\boldsymbol{p}^{(t+1)} - \boldsymbol{p}^{(t)}\right)^\top \nabla^2\mathcal{L}_2(\boldsymbol{p}^{(t)})\left(\boldsymbol{p}^{(t+1)} - \boldsymbol{p}^{(t)}\right) \tag{18}$$

$$\stackrel{\text{(a)}}{=} \mathcal{L}_2(\boldsymbol{p}^{(t)}) + \boldsymbol{g}^\top\left(\boldsymbol{p}^{(t+1)} - \boldsymbol{p}^{(t)}\right) + \frac{1}{2}\left(\boldsymbol{p}^{(t+1)} - \boldsymbol{p}^{(t)}\right)^\top \boldsymbol{A}\left(\boldsymbol{p}^{(t+1)} - \boldsymbol{p}^{(t)}\right) \tag{19}$$

$$\stackrel{\text{(b)}}{=} \mathcal{L}_2(\boldsymbol{p}^{(t)}) + \frac{1}{t+1}\boldsymbol{g}^\top\left(\boldsymbol{e}^{(i^{(t)})} - \boldsymbol{p}^{(t)}\right) + \frac{1}{2(t+1)^2}\left(\boldsymbol{e}^{(i^{(t)})} - \boldsymbol{p}^{(t)}\right)^\top \boldsymbol{A}\left(\boldsymbol{e}^{(i^{(t)})} - \boldsymbol{p}^{(t)}\right) \tag{20}$$

$$\stackrel{\text{(c)}}{\leq} \mathcal{L}_2(\boldsymbol{p}^{(t)}) + \frac{1}{t+1}\boldsymbol{g}^\top\left(\boldsymbol{e}^{(i^{(t)})} - \boldsymbol{p}^{(t)}\right) + \frac{2}{(t+1)^2}, \tag{21}$$

where we define $\boldsymbol{g} = \nabla\mathcal{L}_2(\boldsymbol{p}^{(t)}) = \boldsymbol{A}\boldsymbol{p}^{(t)}$ in (a), (b) follows from Eq. 17, and (c) follows from Assumption 2.4, i.e., $\|\boldsymbol{A}\|_{\max} = \max_{ij}|\boldsymbol{A}_{ij}| \leq 1$, and $\|\boldsymbol{e}^{(i^{(t)})} - \boldsymbol{p}^{(t)}\|_1 \leq 2$. Let $i^* = \arg\min_i \boldsymbol{g}_i$, or equivalently, $\boldsymbol{e}^{(i^*)} = \arg\min_{\boldsymbol{p} \in \Delta_K} \boldsymbol{g}^T \boldsymbol{p}$, i.e., $\boldsymbol{e}^{(i^*)}$ is the direction of the Frank-Wolfe update. From convexity of $L_2$,

$$\boldsymbol{g}^\top(\boldsymbol{e}^{(i^*)} - \boldsymbol{p}^{(t)}) = \min_{\boldsymbol{p} \in \Delta_K} \boldsymbol{g}^\top(\boldsymbol{p} - \boldsymbol{p}^{(t)}) \tag{22}$$

$$= \min_{\boldsymbol{p} \in \Delta_K}\left(\mathcal{L}_2(\boldsymbol{p}^{(t)}) + \boldsymbol{g}^\top(\boldsymbol{p} - \boldsymbol{p}^{(t)})\right) - \mathcal{L}_2(\boldsymbol{p}^{(t)}) \tag{23}$$

$$\leq \mathcal{L}_2(\boldsymbol{p}^*) - \mathcal{L}_2(\boldsymbol{p}^{(t)}). \tag{24}$$

From the definition of $\boldsymbol{e}^{(i^{(t)})}$, we have for $t \geq (K+1)$,

$$\boldsymbol{g}^\top \boldsymbol{e}^{(i^{(t)})} \stackrel{\text{(a)}}{=} \left(\boldsymbol{A}\boldsymbol{p}^{(t)}\right)^\top \boldsymbol{e}^{(i^{(t)})} \tag{25}$$

$$= \frac{1}{t}\left(\boldsymbol{A}\boldsymbol{x}^{(t)}\right)^\top \boldsymbol{e}^{(i^{(t)})} \tag{26}$$

$$= \frac{1}{t}\left(\boldsymbol{l}^{(1)} + \boldsymbol{A}\boldsymbol{x}^{(t)}\right)^{\top}\boldsymbol{e}^{(i^{(t)})} - \frac{1}{t}\boldsymbol{l}^{(1)\top}\boldsymbol{e}^{(i^{(t)})} \tag{27}$$

$$\leq \frac{1}{t}\left(\boldsymbol{l}^{(1)} + \boldsymbol{A}\boldsymbol{x}^{(t)}\right)^{\top}\boldsymbol{e}^{(i^{(t)})} + \frac{\|\boldsymbol{l}^{(1)}\|_{\infty}}{t} \tag{28}$$

$$= \frac{1}{t}\boldsymbol{l}^{(t)\top}\boldsymbol{e}^{(i^{(t)})} + \frac{\|\boldsymbol{l}^{(1)}\|_{\infty}}{t} \tag{29}$$

$$= \frac{1}{t}l^{(t)}_{i^{(t)}} + \frac{\|\boldsymbol{l}^{(1)}\|_{\infty}}{t} \tag{30}$$

$$\overset{(b)}{\leq} \frac{1}{t}\left(\hat{l}^{(t)}_{i^{(t)}} + c^{(t)}_{i^{(t)}} + 1\right) + \frac{\|\boldsymbol{l}^{(1)}\|_{\infty}}{t} \tag{31}$$

$$\overset{(c)}{\leq} \frac{1}{t}\left(\hat{l}^{(t)}_{i^*} + 2c^{(t)}_{i^{(t)}} - c^{(t)}_{i^*} + 1\right) + \frac{\|\boldsymbol{l}^{(1)}\|_{\infty}}{t} \tag{32}$$

$$\leq \frac{1}{t}\left(l^{(t)}_{i^*} + 2c^{(t)}_{i^{(t)}} + 2\right) + \frac{\|\boldsymbol{l}^{(1)}\|_{\infty}}{t} \tag{33}$$

$$= \frac{1}{t}\left(\left(\boldsymbol{l}^{(1)} + \boldsymbol{A}\boldsymbol{x}^{(t)}\right)^{\top}\boldsymbol{e}^{(i^*)} + 2c^{(t)}_{i^{(t)}} + 2\right) + \frac{\|\boldsymbol{l}^{(1)}\|_{\infty}}{t} \tag{34}$$

$$\leq \frac{1}{t}\left(\left(\boldsymbol{A}\boldsymbol{x}^{(t)}\right)^{\top}\boldsymbol{e}^{(i^*)} + 2c^{(t)}_{i^{(t)}} + 2\right) + \frac{2\|\boldsymbol{l}^{(1)}\|_{\infty}}{t} \tag{35}$$

$$= \boldsymbol{g}^{\top}\boldsymbol{e}^{(i^*)} + \frac{1}{t}\left(2c^{(t)}_{i^{(t)}} + 2\right) + \frac{2\|\boldsymbol{l}^{(1)}\|_{\infty}}{t} \tag{36}$$

$$= \boldsymbol{g}^{\top}\boldsymbol{e}^{(i^*)} + \frac{2}{t}\left(c^{(t)}_{i^{(t)}} + 1 + \|\boldsymbol{l}^{(1)}\|_{\infty}\right), \tag{37}$$

where (a) follows from the definition of $\boldsymbol{g}$, (b) follows from $(\hat{l}^{(t)}_{i^{(t)}} + c^{(t)}_{i^{(t)}} + 1)$ is an upper bound of the expected loss because it increases at most $c^{(t)}_{i^{(t)}}$ from the previous observation, the noise is bounded by 1, and $\hat{l}^{(t)}_i$ is set to be the previous observation for $t \geq (K+1)$, and (c) follows from the algorithm chooses $i^{(t)}$ such that $\hat{l}^{(t)}_{i^{(t)}} - c^{(t)}_{i^{(t)}} \leq \hat{l}^{(t)}_i - c^{(t)}_i$ for all $i \in [K]$. By combining Eqs. 21, 24, and 37, we have

$$\delta(t+1) \overset{\text{def}}{=} \mathcal{L}_2(\boldsymbol{p}^{(t+1)}) - \mathcal{L}_2(\boldsymbol{p}^*) \tag{38}$$

$$\overset{(a)}{\leq} \mathcal{L}_2(\boldsymbol{p}^{(t)}) + \frac{1}{t+1}\boldsymbol{g}\left(\boldsymbol{e}^{(i^{(t)})} - \boldsymbol{p}^{(t)}\right) + \frac{2}{(t+1)^2} - \mathcal{L}_2(\boldsymbol{p}^*) \tag{39}$$

$$\overset{(b)}{\leq} \mathcal{L}_2(\boldsymbol{p}^{(t)}) + \frac{1}{t+1}\boldsymbol{g}\left(\boldsymbol{e}^{(i^*)} - \boldsymbol{p}^{(t)}\right) + \frac{2}{(t+1)^2} + \frac{2}{t(t+1)}\left(c^{(t)}_{i^{(t)}} + 1 + \|\boldsymbol{l}^{(1)}\|_{\infty}\right) - \mathcal{L}_2(\boldsymbol{p}^*) \tag{40}$$

$$\overset{(c)}{\leq} \mathcal{L}_2(\boldsymbol{p}^{(t)}) + \frac{1}{t+1}\left(\mathcal{L}_2(\boldsymbol{p}^*) - \mathcal{L}_2(\boldsymbol{p}^{(t)})\right) + \frac{2}{(t+1)^2} + \frac{2}{t(t+1)}\left(c^{(t)}_{i^{(t)}} + 1 + \|\boldsymbol{l}^{(1)}\|_{\infty}\right) - \mathcal{L}_2(\boldsymbol{p}^*) \tag{41}$$

$$= \frac{t}{t+1}\left(\mathcal{L}_2(\boldsymbol{p}^{(t)}) - \mathcal{L}_2(\boldsymbol{p}^*)\right) + \frac{2}{(t+1)^2} + \frac{2}{t(t+1)}\left(c^{(t)}_{i^{(t)}} + 1 + \|\boldsymbol{l}^{(1)}\|_{\infty}\right) \tag{42}$$

$$= \frac{t}{t+1}\delta(t) + \frac{2}{(t+1)^2} + \frac{2}{t(t+1)}\left(c^{(t)}_{i^{(t)}} + 1 + \|\boldsymbol{l}^{(1)}\|_{\infty}\right), \tag{43}$$

where (a) follows from Eq. 21, (b) follows from Eq. 37, and (c) follows from Eq. 24. By multiplying $t+1$ to both sides and letting $\tilde{\delta}(t) = t\delta(t)$, and recursively applying it from $t = (K+1)$ to $T-1$,

$$\tilde{\delta}(T) \leq \tilde{\delta}(T-1) + \frac{2}{T} + \frac{2}{T-1}\left(c^{(T-1)}_{i^{(T-1)}} + 1 + \|\boldsymbol{l}^{(1)}\|_{\infty}\right) \tag{44}$$

$$\leq \tilde{\delta}(K+1) + \sum_{s=K+1}^{T-1} \frac{2}{(s+1)} + 2\sum_{s=K+1}^{T-1} \frac{c_{i(s)}^{(s)} + 1 + \|\boldsymbol{l}^{(1)}\|_\infty}{s} \tag{45}$$

Let $\Phi(1) = 0$ and $\Phi(t+1) = \Phi(t) + \frac{K}{t}$. We show $\sum_{s=1}^{t-1} \frac{c_{i(s)}^{(s)}}{s} + \frac{1}{t-1}\sum_{i\in[K]} c_i^{(t)} \leq \Phi(t)$ for $t \geq 2$ by induction. When $t = 2$,

$$\sum_{s=1}^{t-1} \frac{c_{i(s)}^{(s)}}{s} + \frac{1}{t-1}\sum_{i\in[K]} c_i^{(t)} = \frac{c_{i(1)}^{(1)}}{1} + \frac{1}{1}\sum_{i\in[K]} c_i^{(2)} \tag{46}$$

$$\overset{(a)}{=} 0 + \sum_{i\in[K]} c_i^{(2)} \tag{47}$$

$$\overset{(b)}{=} K \tag{48}$$

$$= \Phi(2), \tag{49}$$

where (a) follows from $c_i^{(1)} = 0$, and (b) follows from $c_i^{(1)}$ is incremented by 1. Suppose the induction hypothesis $\sum_{s=1}^{t-1} \frac{c_{i(s)}^{(s)}}{s} + \frac{1}{t-1}\sum_{i\in[K]} c_i^{(t)} \leq \Phi(t)$ holds. Then,

$$\sum_{s=1}^{t} \frac{c_{i(s)}^{(s)}}{s} + \frac{1}{t}\sum_{i\in[K]} c_i^{(t+1)} \overset{(a)}{=} \sum_{s=1}^{t} \frac{c_{i(s)}^{(s)}}{s} + \frac{1}{t}\left(1 + \sum_{i\neq i^{(t)}} c_i^{(t+1)}\right) \tag{50}$$

$$= \sum_{s=1}^{t} \frac{c_{i(s)}^{(s)}}{s} + \frac{1}{t}\left(1 + \sum_{i\in[K]} c_i^{(t)} + (K-1) - c_{i^{(t)}}^{(t)}\right) \tag{51}$$

$$= \sum_{s=1}^{t} \frac{c_{i(s)}^{(s)}}{s} + \frac{1}{t}\sum_{i\in[K]} c_i^{(t)} + \frac{K}{t} - \frac{c_{i^{(t)}}^{(t)}}{t} \tag{52}$$

$$= \sum_{s=1}^{t-1} \frac{c_{i(s)}^{(s)}}{s} + \frac{1}{t}\sum_{i\in[K]} c_i^{(t)} + \frac{K}{t} \tag{53}$$

$$\leq \sum_{s=1}^{t-1} \frac{c_{i(s)}^{(s)}}{s} + \frac{1}{t-1}\sum_{i\in[K]} c_i^{(t)} + \frac{K}{t} \tag{54}$$

$$\overset{(b)}{\leq} \Phi(t) + \frac{K}{t} \tag{55}$$

$$= \Phi(t+1). \tag{56}$$

where (a) follows from the fact that the algorithm increments $c_i^{(t)}$ by 1 and resets $c_{i^{(t)}}^{(t)}$ to 1, and (b) follows from the induction hypothesis. Since $\Phi(T) = K\sum_{t=1}^{T-1} \frac{1}{t} \leq K(1+\log T)$ and $c_i^{(t)} \geq 0$, we have $\sum_{s=1}^{T-1} \frac{c_{i(s)}^{(s)}}{s} \leq K(1+\log T)$. Therefore,

$$\tilde{\delta}(T) \leq \tilde{\delta}(K+1) + \sum_{s=K+1}^{T-1} \frac{2}{(s+1)} + 2\sum_{s=K+1}^{T-1} \frac{c_{i(s)}^{(s)} + 1 + \|\boldsymbol{l}^{(1)}\|_\infty}{s} \tag{57}$$

$$\leq \tilde{\delta}(K+1) + 2\log T + 2K(1+\log T) + 2\log T + 2\|\boldsymbol{l}^{(1)}\|_\infty \log T \tag{58}$$

$$\leq \tilde{\delta}(K+1) + 2K + \left(2K + 2\|\boldsymbol{l}^{(1)}\|_\infty + 4\right)\log T \tag{59}$$

$$= (K+1)\delta(K+1) + 2K + \left(2K + 2\|\boldsymbol{l}^{(1)}\|_\infty + 4\right)\log T \tag{60}$$

$$\leq (K+1)\left(\mathcal{L}_2(\boldsymbol{p}^{(K+1)}) - \mathcal{L}_2(\boldsymbol{p}^*)\right) + 2K + \left(2 + 2\|\boldsymbol{l}^{(1)}\|_\infty + 4\right)\log T \tag{61}$$

$$\overset{(a)}{\leq} (K+1)\mathcal{L}_2(\boldsymbol{p}^{(K+1)}) + 2K + \left(2K + 2\|\boldsymbol{l}^{(1)}\|_\infty + 4\right)\log T \tag{62}$$

$$= \frac{K+1}{2}\boldsymbol{p}^{(K+1)\top}\boldsymbol{A}\boldsymbol{p}^{(K+1)} + 2K + \left(2K + 2\|\boldsymbol{l}^{(1)}\|_\infty + 4\right)\log T \tag{63}$$

$$\leq \frac{K+1}{2} + 2K + \left(2K + 2\|\boldsymbol{l}^{(1)}\|_\infty + 4\right)\log T \tag{64}$$

$$= \frac{5K+1}{2} + \left(2K + 2\|\boldsymbol{l}^{(1)}\|_\infty + 4\right)\log T, \tag{65}$$

where (a) follows from $\boldsymbol{A}$ is positive semi-definite and $\mathcal{L}_2(\boldsymbol{p}^*) \geq 0$. Since $\tilde{\delta}(T) = T\delta(T)$,

$$\mathcal{L}_2(\boldsymbol{p}^{(T)}) - \mathcal{L}_2(\boldsymbol{p}^*) \leq \frac{5K+1}{2T} + \left(2K + 2\|\boldsymbol{l}^{(1)}\|_\infty + 4\right)\frac{\log T}{T}. \tag{66}$$

Therefore, the regret is

$$\mathcal{L}(\boldsymbol{x}^{(T)}) - \mathcal{L}(\boldsymbol{x}^{(T)*}) = \boldsymbol{b}^\top\boldsymbol{x}^{(T)} + \frac{1}{2}\boldsymbol{x}^{(T)\top}\boldsymbol{A}\boldsymbol{x}^{(T)} - \boldsymbol{b}^\top\boldsymbol{x}^{(T)*} - \frac{1}{2}\boldsymbol{x}^{(T)*\top}\boldsymbol{A}\boldsymbol{x}^{(T)*} \tag{67}$$

$$\leq \frac{1}{2}\boldsymbol{x}^{(T)\top}\boldsymbol{A}\boldsymbol{x}^{(T)} - \frac{1}{2}\boldsymbol{x}^{(T)*\top}\boldsymbol{A}\boldsymbol{x}^{(T)*} + 2T\|\boldsymbol{b}\|_\infty \tag{68}$$

$$= \frac{T^2}{2}\boldsymbol{p}^{(T)\top}\boldsymbol{A}\boldsymbol{p}^{(T)} - \frac{T^2}{2}\frac{\boldsymbol{x}^{(T)*\top}}{T}\boldsymbol{A}\frac{\boldsymbol{x}^{(T)*}}{T} + 2T\|\boldsymbol{b}\|_\infty \tag{69}$$

$$\overset{(a)}{\leq} \frac{T^2}{2}\boldsymbol{p}^{(T)\top}\boldsymbol{A}\boldsymbol{p}^{(T)} - \frac{T^2}{2}\boldsymbol{p}^{*\top}\boldsymbol{A}\boldsymbol{p}^* + 2T\|\boldsymbol{b}\|_\infty \tag{70}$$

$$= T^2(\mathcal{L}_2(\boldsymbol{p}^{(T)}) - \mathcal{L}_2(\boldsymbol{p}^*)) + 2T\|\boldsymbol{b}\|_\infty \tag{71}$$

$$\leq \frac{(5K+1)T}{2} + \left(2K + 2\|\boldsymbol{l}^{(1)}\|_\infty + 4\right)T\log T + 2T\|\boldsymbol{b}\|_\infty \tag{72}$$

$$\overset{(b)}{\leq} \frac{(5K+1)T}{2} + \left(2K + 2\|\boldsymbol{l}^{(1)}\|_\infty + 4\right)T\log T + 2T\left(\|\boldsymbol{l}^{(1)}\|_\infty + \frac{1}{2}\right) \tag{73}$$

$$= \left(\frac{5K+3}{2} + 2\|\boldsymbol{l}^{(1)}\|_\infty\right)T + \left(2K + 2\|\boldsymbol{l}^{(1)}\|_\infty + 4\right)T\log T \tag{74}$$

$$= O(KT\log T), \tag{75}$$

where (a) follows from $\frac{\boldsymbol{x}^{(T)*\top}}{T} \in \Delta_K$, and (b) follows from $\boldsymbol{b} = \boldsymbol{l}^{(1)} - \frac{1}{2}[\boldsymbol{A}_{11}, \boldsymbol{A}_{22}, \ldots, \boldsymbol{A}_{KK}]^\top$. $\qquad\square$

We believe that this analysis is of independent interest. The gradient of the objective function is $\nabla\mathcal{L}(\boldsymbol{x}^{(t)}) = \boldsymbol{b} + \boldsymbol{A}\boldsymbol{x}^{(t)} = \boldsymbol{l}^{(t)} + \text{Const}$. When we apply the Frank-Wolfe algorithm[8, 16, 22] to this problem, the update is the form of $\boldsymbol{x}^{(t+1)} \leftarrow \alpha\boldsymbol{x}^{(t)} + \beta\boldsymbol{e}^{(i^{(t)}_{\text{Frank-Wolf}})}$, where $\boldsymbol{e}^{(i^{(t)}_{\text{Frank-Wolf}})} = \arg\min_{\boldsymbol{p}\in\Delta_K}\nabla\mathcal{L}(\boldsymbol{x}^{(t)})^\top\boldsymbol{p}$ is the Frank-Wolfe update, and since the optimization domain is the simplex and the linearized function achieves the minimum at the corner of the simplex, $i^{(t)}_{\text{Frank-Wolf}} = \arg\min_i \left(\nabla\mathcal{L}(\boldsymbol{x}^{(t)})\right)_i = \arg\min_i \left(\boldsymbol{b} + \boldsymbol{A}\boldsymbol{x}^{(t)}\right)_i = \arg\min_i \left(\boldsymbol{l}^{(t)} + \text{Const}\right)_i$ is the arm that minimizes the gradient. The key insight is here.



The Frank-Wolfe update on the simplex domain

= Selection of one corner of the simplex

= Selection of one arm.



Therefore, the Frank-Wolf algorithm sequentially chooses the arm that minimizes the gradient (or the current loss) and inserts it into the current state. This behavior is similar to the influential LCB algorithm. We exploited this connection and applied the convergence analysis of the Frank-Wolfe algorithm to the influential LCB algorithm.

**Remark (Assumption on the Norm of $A$).** The assumption $\|A\|_{\max} = \max_{ij} |A_{ij}| \leq 1$ is not necessary for the influential LCB algorithm and the regret analysis. Alternatively, if we know the scale of rewards in advance, we can normalize the rewards accordingly. When we know that $\|A\|_{\max} \leq B$, a lower bound of the loss is $\hat{l}_i - Bc_i$, and the influential LCB algorithm chooses the arm that minimizes the lower bound in Line 4. When we have no prior knowledge of the norm of $A$, it can be estimated in the initial phase of the algorithm. If $i^{(t)} = i^{(t+1)} = j$, $L^{(t)} = l_j^{(t)} + \xi^{(t)}$ and $L^{(t+1)} = l_j^{(t+1)} + \xi^{(t+1)} = l_j^{(t)} + A_{jj} + \xi^{(t+1)}$. Thus, $\hat{A}_{jj} \overset{\text{def}}{=} L^{(t+1)} - L^{(t)} = A_{jj} + \xi^{(t+1)} - \xi^{(t)}$ is a noisy estimate of $A_{jj}$ with at most a constant error. If $i^{(t)} = i^{(t+2)} = j$ and $i^{(t+1)} = k$, $\hat{A}_{jk} \overset{\text{def}}{=} L^{(t+2)} - L^{(t)} - \hat{A}_{jj} = A_{jj} + A_{jk} + \xi^{(t+2)} - \xi^{(t)} - \hat{A}_{jj}$ is a noisy estimate of $A_{jk}$ with at most a constant error. Therefore, with $O(K^2)$ samples, we can estimate $A$ with a constant error. The same analysis as Theorem 2.6 is valid for these cases.

## 3 Related Work

The *restless bandit* [25, 29], *rotting bandit* [21, 24], and *rising bandit problems* [14] are particularly closely related to this work. In these settings, each arm is associated with a state that evolves depending on whether the arm is selected, and the reward distribution varies according to the current state. In the rotting bandit model, it is assumed that the reward of a selected arm decays over time, whereas in the rising bandit, the reward increases. The restless bandit further generalizes this setting by allowing the states and hence the rewards of unselected arms to evolve according to a different dynamic. However, a key distinction from our model is that in these frameworks, the state of each arm depends only on its own selection history. In contrast, the influential bandit problem considers interdependencies, where selecting one arm can directly affect the future losses of other arms. On the other hand, those prior models allow for non-linear reward dynamics, so our setting is not necessarily more general in all aspects. Notably, the recently proposed *graph-triggered rising bandit* problem [9] also models interactions across arms, where selecting one arm can influence the state of others. While similar in spirit, it differs from our model in assuming monotonic non-decreasing rewards and restricting interactions based on an underlying graph structure. Our model supports general interactions modeled by an interaction matrix and can handle both increases and decreases in rewards.

Our model can also be interpreted within the framework of a Markov Decision Process (MDP) or a Partially Observable MDP (POMDP). For example, if we define the system state at each time step as the vector of selection counts for all arms, then the state transitions deterministically according to the chosen arm, and the reward can be modeled as a (possibly unknown) stochastic function of the state and selected action. In this sense, the problem can be viewed as an instance of an MDP. One might therefore consider applying online learning algorithms designed for MDPs. However, to the best of our knowledge, existing algorithms are not directly applicable to our setting. Most online MDP algorithms have been developed for the *episodic setting*, where learning occurs over finite-horizon episodes that restart from an initial state, such as in [3, 4, 15, 17, 18]. This setting differs significantly from ours, which operates in a non-episodic, continual learning framework. Although there are some studies on the *infinite-horizon average reward model* [2, 11, 27, 28], these typically rely on assumptions such as bounded state spaces or ergodicity/mixing conditions on the state transitions, which do not hold in our setting. Online control [10, 13] is also a general framework capable of handling state transitions and uncertainty; however, approaches in this research area typically focus on continuous variables and often rely on assumptions such as stabilizability or controllability, making them not directly applicable to our problem setting.

## 4 Experiments

We validate the effectiveness of influential bandits through numerical experiments.

### 4.1 Regret Analysis

We confirm that the regret of Algorithm 1 is near-linear (Theorem 2.6) while the standard LCB algorithm incurs a near-quadratic regret (Proposition 2.2).

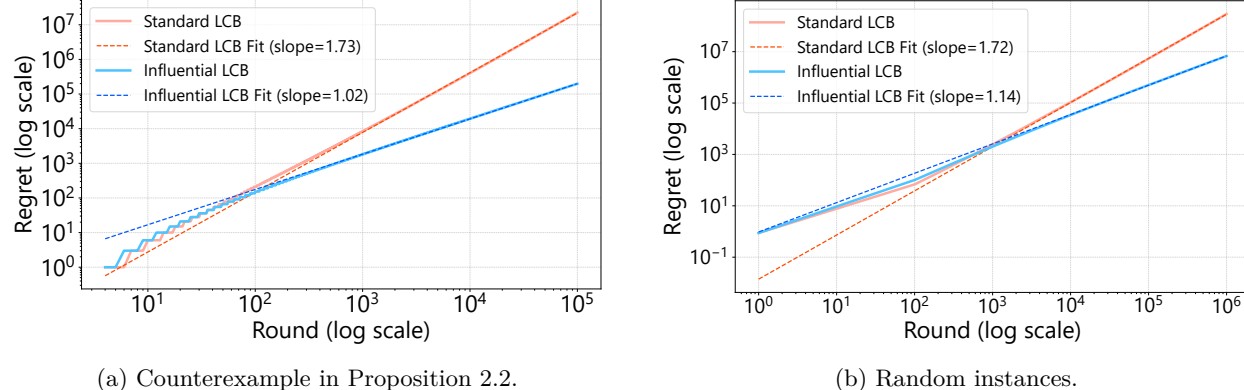

(a) Counterexample in Proposition 2.2.

(b) Random instances.

Figure 1: **Regret Analysis.** The x-axis is the time horizon $T$ in log scale, and the y-axis is the regret in log scale. We fit linear regression lines to the log-log plot. (a) The slope of the fit line is 1.02 for the influential LCB algorithm and 1.73 for the standard LCB algorithm. The former converges to 1 and the latter converges to 2 as $T$ increases. This validates that the standard LCB algorithm incurs a near-quadratic regret in the worst case. (b) The slope of the fit line is 1.14 for the influential LCB algorithm and 1.72 for the standard LCB algorithm. This case also shows that the rate of the regret of the influential LCB algorithm is kept near-linear while the standard LCB algorithm can be large even in the average case.

We use two settings. The first setting is the same as the counterexample in Proposition 2.2, i.e., $K = 2, \boldsymbol{A} = [[1,1],[1,2]], \boldsymbol{l}^{(1)} = [1,1]^\top, \xi^{(t)} = 0$. The optimal sequence of actions is $i^{(t)} = 1$ for all $t \in [T]$, and the total loss is $\frac{1}{2}T(T+1)$. Figure 1 (a) shows that the growth rate of the regret of the influential LCB algorithm is near-linear, while the standard LCB algorithm incurs a near-quadratic regret.

The second setting uses random instances with $K = 3$ arms. We generate random semi-positive definite matrices $\boldsymbol{A}$ by sampling the entries of $\boldsymbol{B} \in \mathbb{R}^{K \times K}$ from the standard normal distribution and setting $\boldsymbol{A} = \frac{\boldsymbol{B}^\top \boldsymbol{B}}{\max_{ij}|\boldsymbol{B}^\top \boldsymbol{B}|_{ij}}$. $\boldsymbol{l}^{(1)} \in \mathbb{R}^K$ is sampled from the standard normal distribution. The noise $\xi_i^{(t)}$ is sampled from the standard normal distribution. Since the optimal sequence of actions is unknown, we use the continuous relaxation $L^* = \min_{\boldsymbol{x} \in t \Delta_K} \boldsymbol{b}^\top \boldsymbol{x} + \frac{1}{2}\boldsymbol{x}^\top \boldsymbol{A} \boldsymbol{x}$ as the benchmark, which is a provable lower bound of the expected total loss. We define the difference between the expected loss $\sum_{t=1}^T \boldsymbol{l}_{i^{(t)}}^{(t)}$ of algorithm choices $i^{(1)}, \ldots, i^{(T)}$ and the benchmark score $L^*$ as the regret. We run the experiments with 100 seeds and Figure 1 (b) reports the average regret. The result shows that the growth rate of the regret of the influential LCB algorithm is kept near-linear, while the standard LCB algorithm incurs a near-quadratic regret even in random instances.

## 4.2 Model Analysis

We confirm that pulling an arm can indeed change the losses of arms and validate the model of influential bandits with real data. We use MovieLens-32M dataset [12]. We view each user as solving a multi-armed bandit problem, treating each user as an independent instance. A user selects a movie, watches it, and the user's preference is revealed as the loss. Since there are too many movies, and each user selects a movie at most once, we consider a movie genre as an arm rather than an individual movie. Therefore $K = \#\text{genres} = 20$. Since a movie can have multiple genres, we randomly select a genre of a movie and assign it to the movie. We define the loss as $(5 - \text{rating})$ since the 5-star rating is recorded in this dataset. The history of genre selections and ratings defines log data of a bandit problem.

To examine whether the assumption of stationary losses or the assumption of influential bandits is more appropriate, we evaluate how well each model predicts future losses. We use a leave-one-out validation approach. Specifically, we sort each user's history in chronological order and fit the model using all but the last rating. The final rating is then predicted using the trained model.

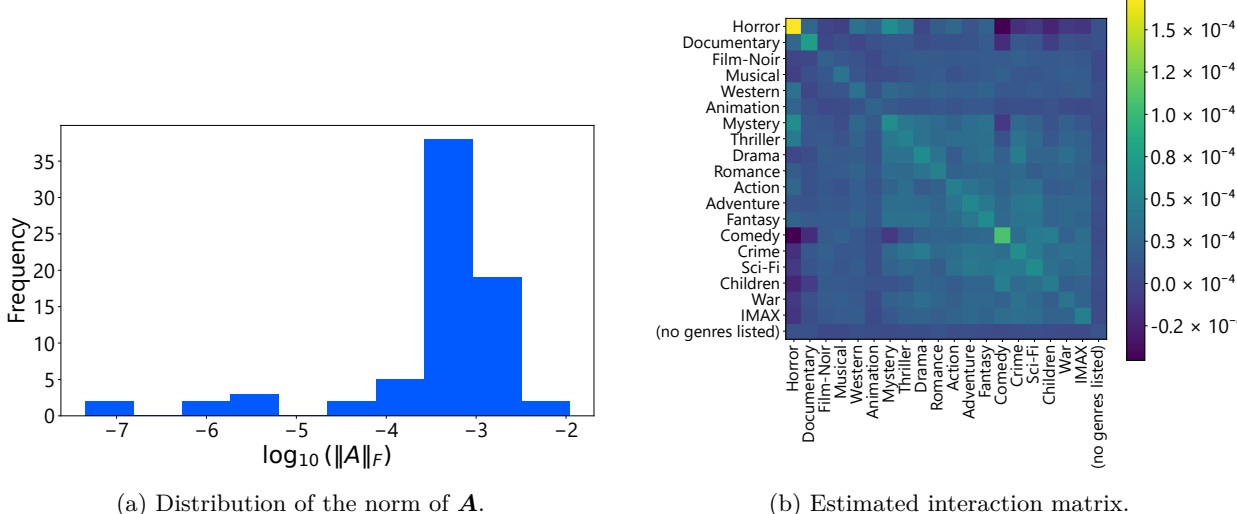

(a) Distribution of the norm of $\boldsymbol{A}$.

(b) Estimated interaction matrix.

Figure 2: **Model Analysis.** (a) The distribution of the norm of $\boldsymbol{A}$. The majority of users have a norm greater than $10^{-4}$, which implies the existence of interaction among arms. However, there are some users with almost zero norms, which indicates that the positive semi-definite interaction is not valid for these users. (b) The averaged interaction matrix $\bar{\boldsymbol{A}}$ as a heatmap. The diagonal elements, such as horror and comedy, are high, which indicates that consecutively watching similar movies increases fatigue. The off-diagonal elements, such as those between horror and comedy, are negative, indicating that watching a comedy movie helps recover from the fatigue caused by watching a horror movie.

Under the assumption of stationary losses, we estimate the mean loss for each genre (arm) based on the training data and predict the loss of the last chosen genre using this mean value. In contrast, the influential bandits model assumes that $\boldsymbol{l}^{(t)} \approx \boldsymbol{l}^{(1)} + \boldsymbol{A}\boldsymbol{x}^{(t)}$, and we fit $\boldsymbol{l}^{(1)} \in \mathbb{R}^K$ and $\boldsymbol{A} \in \mathbb{R}^{K \times K}$ to the data. Since $\boldsymbol{A}$ is positive semi-definite, we rewrite the model as $\boldsymbol{l}^{(t)} \approx \boldsymbol{l}^{(1)} + \boldsymbol{B}\boldsymbol{B}^T\boldsymbol{x}^{(t)}$ and optimize $\boldsymbol{l}^{(1)}$ and $\boldsymbol{B} \in \mathbb{R}^{K \times K}$ as parameters. The number of parameters to estimate is thus $(K + K^2) = 420$. We minimize the squared error between the observed values and the predicted values using gradient descent with momentum. Finally, we use the obtained $\boldsymbol{l}^{(1)}$ and $\boldsymbol{A}$ to predict the loss for the last selection.

To ensure sufficient training data, we consider only users with at least 4096 data points, resulting in $n = 73$ valid users. We conducted experiments for each user. The mean squared error (MSE) for predicting the final loss was $0.6156 \pm 0.9775$ under the stationary loss assumption and $0.5720 \pm 0.8238$ under the influential bandits assumption. The influential bandits model achieved better predictive accuracy, suggesting that losses are not stationary. It should be noted that the difference is not statistically significant since the ratings are highly noisy to predict.

To further examine the validity of the assumptions, we analyze the estimated $\boldsymbol{A}$. First, Figure 2 (a) presents the distribution of the norms of the estimated $\boldsymbol{A}$. Each sample corresponds to a single user. The majority of users have a norm greater than $10^{-4}$, and there are 20 users with a norm exceeding $10^{-3}$. This indicates that some users exhibit strong interactions. On the other hand, 7 users have norms below $10^{-5}$, and the estimated interaction matrix $\boldsymbol{A}$ is almost zero, suggesting that for these users, the assumed positive-definite interaction, where watching similar movies increases loss (i.e., "fatigue"), is not observed. Instead, there may even be a "reinforcement" effect where watching similar movies reduces loss. We further discuss this phenomenon in Section 5.1. These findings suggest that the validity of the influential bandits assumption depends on the user and the environment, but there are indeed cases where the influential bandits model is reasonable.

Next, we visualize the averaged interaction matrix $\bar{\boldsymbol{A}}$ as a heatmap in Figure 2 (b). Notably, $\boldsymbol{A}_{\text{Horror,Horror}}$ and $\boldsymbol{A}_{\text{Comedy,Comedy}}$ exhibit significantly high values. This suggests that watching horror movies consecutively or comedy movies consecutively leads to increased losses (i.e., decreased ratings). Additionally, $\boldsymbol{A}_{\text{Horror,Comedy}}$

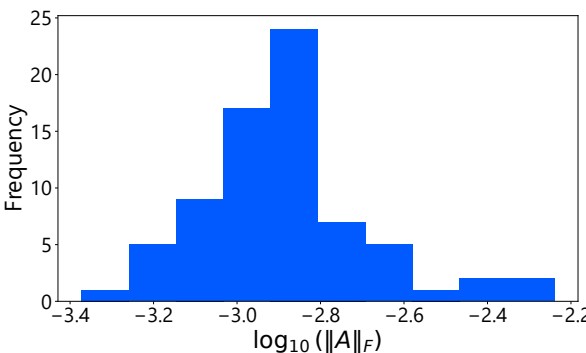

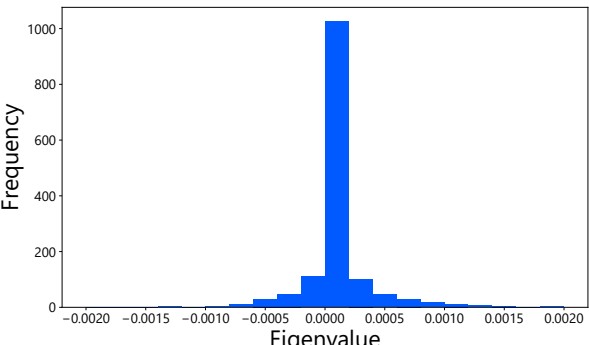

(a) Distribution of the norm of $\boldsymbol{A}$ when $\boldsymbol{A}$ is not required to be positive semi-definite.

(b) Distribution of the eigenvalues of $\boldsymbol{A}$ when $\boldsymbol{A}$ is not required to be positive semi-definite.

Figure 3: **indefinite Version.** (a) The distribution of the norms of $\boldsymbol{A}$ when $\boldsymbol{A}$ is not required to be positive semi-definite. All users have norms greater than $10^{-4}$. This suggests that negative eigenvalues are required to model some users. (b) The distribution of the eigenvalues of $\boldsymbol{A}$. More than half of the eigenvalues are positive, which validates our positive semi-definite and "fatigue" assumption, but there are many negative eigenvalues as well. This suggests that there are "reinforcement" effects where watching similar movies reduces loss by, e.g., gaining further knowledge and interest in the genre.

is negative, implying that watching a comedy movie reduces the loss of horror movies, suggesting that watching a horror movie after a comedy movie enhances enjoyment. Furthermore, we observe weak positive clusters among {Action, Adventure, Fantasy} and {Mystery, Thriller, Drama, Romance}, indicating that watching one of these genres increases fatigue toward other similar genres as well. These effects align with intuitive expectations and support the validity of the influential bandits model.

## 5 Discussions

### 5.1 Limitation: Positive Semi-Definite Interaction Matrix

We have assumed that the interaction matrix $\boldsymbol{A}$ is positive semi-definite. This is reasonable because repeating the same action increases the loss in many scenarios, and we partially validated this assumption in Section 4.2. However, this assumption may not always hold as we observed in Section 4.2. We conducted the experiments with MovieLens-32M dataset without the positive semi-definite constraint on $\boldsymbol{A}$. Specifically, we parameterize $\boldsymbol{A}$ as $\boldsymbol{A} = \boldsymbol{M} + \boldsymbol{M}^\top$ where $\boldsymbol{M} \in \mathbb{R}^{K \times K}$ is a free parameter and fit $\boldsymbol{l}^{(1)}$ and $\boldsymbol{M}$ to the data with the same leave-one-out validation approach. The mean squared error (MSE) for predicting the final loss is $0.5655 \pm 0.8119$, which is slightly better than the positive semi-definite version. Figure 3 (a) shows the distribution of the norms of $\boldsymbol{A}$. Whereas some users have almost zero norms in the positive semi-definite version, all users have norms greater than $10^{-4}$ in the indefinite version. This suggests that negative eigenvalues are required to model some users. Figure 3 (b) shows the distribution of the eigenvalues of $\boldsymbol{A}$. There are #users $\times K$ eigenvalues. More than half of the eigenvalues are positive, which validate our positive semi-definite and "fatigue" assumption, but there are many negative eigenvalues as well. This also suggests that there are "reinforcement" effects where watching similar movies reduces loss by, e.g., gaining further knowledge and interest in the genre.

Our framework can be applied to this indefinite version as well. However, the challenge is establishing the theoretical guarantee. The positive semi-definite version is framed as an online convex minimization problem. The indefinite version is not convex. The indefinite quadratic optimization problem is NP-hard in general even in the offline case [23]. Therefore, the online bandit case is intractable in general. It is an interesting future work to formulate a tractable setting that can capture the reinforcement effect.

## 5.2 Limitation: Performance in Small Horizons

The primary goal of this paper was to propose a simple algorithm with asymptotically good theoretical performance. The proposed Algorithm 1 is remarkably simple while achieving near-optimal asymptotic performance, as shown in Proposition 2.3 and Theorem 2.6. Thanks to the absence of bells and whistles, it highlights the essence of the proposed influential bandit framework. However, this simplicity may come at the cost of suboptimal performance in finite-time settings. Indeed, in Figure 1 (b), the proposed method underperforms the standard LCB algorithm when $T < 10^3$.

The proposed Algorithm 1 adopts a highly conservative assumption that $\boldsymbol{A}$ may decrease the loss by 1. In reality, $\boldsymbol{A}$ can be estimated with some accuracy using past data. Specifically, as performed in Section 4.2, we can fit $\boldsymbol{l}^{(1)}$ and $\boldsymbol{A}$ to the past arm selection history and observed loss data. Using the estimated $\boldsymbol{A}$, we can obtain a tighter lower bound on the current loss, which is expected to improve performance in finite-time settings. To advance real-world applications, studying techniques to improve finite-time performance, in addition to ensuring asymptotic performance, would be beneficial.

## 5.3 Sublinear Regret

As shown in Proposition 2.3, achieving sublinear regret is not generally possible. In some tasks, however, linear regret may not be acceptable. In the counterexample provided in Proposition 2.3, there exists an arm whose selection leads to irreversible consequences. However, such arms may not exist in practical settings. In reality, an arm's loss may recover over time, making it feasible to select it again. In such cases, the optimal strategy may exhibit a cycle-like pattern, where all arms are pulled at a certain frequency. This implies even bad choices can be made up for in a certain frequency and past mistakes may have only a limited long-term impact, suggesting that sublinear regret might be achievable.

In the experiments on random instances in Section 4.1, the influential LCB method typically exhibited approximately linear regret on average, whereas the standard LCB method suffered approximately quadratic regret. However, in some instances, influential LCB achieved regret lower than the linear rate. We fit a linear regression line to the log-log plot of each random instance and calculated the slope of the line. Figure 4 shows the histogram of the slopes. There are two clusters. The majority of instances (61 out of 100) exhibit a slope close to 1, but 39 instances exhibit a slope close to zero. This suggests that there are "easy" instances where the regret grows sublinearly (e.g., $O(\log T)$). Identifying such cases and designing effective algorithms to exploit them remains an important open problem.

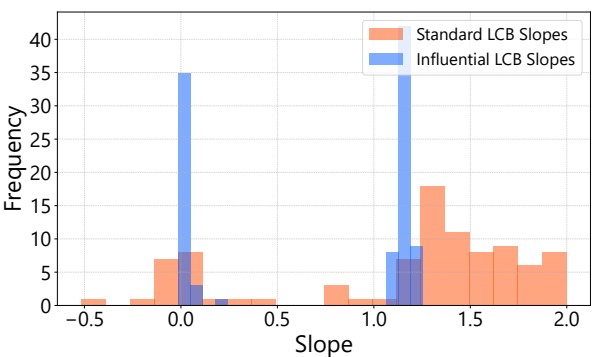

Figure 4: Histogram of the slopes of the regret in random instances.

## 6 Conclusion

In this paper, we made the following contributions.

- We introduced a new problem setting, influential bandits, where the loss of an arm is influenced by the previous selections of all arms (Section 2).

- We showed theoretical properties of influential bandits.
  - We showed that the regret of the standard LCB algorithm is near-quadratic in the worst case (Proposition 2.2).
  - We showed that there exists an instance where the regret of any algorithm is at least linear (Proposition 2.3).

- We proposed a simple algorithm, influential LCB, that achieves near-optimal regret in the worst case (Algorithm 1, Theorem 2.6).

- We empirically validated the influential bandits framework and the influential LCB algorithm through experiments (Sections 4.1 – 4.2).

  - We confirmed that the regret of the proposed algorithm is almost linear in the counterexample and random instances, while the standard LCB algorithm incurs a near-quadratic regret even in the average case (Section 4.1).
  - We confirmed that the loss of an arm can be influenced by the previous selections, and the influential bandits model captures this phenomenon reasonably well with real data (Section 4.2).

We believe that this work introduces a new research direction for handling the influence of past actions in a theoretically sound manner. We hope that our work will inspire further research in this area.

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

## A    Proof of Proposition 2.2

*Proof.* Let $K = 2, \boldsymbol{A} = [[1,1],[1,2]], \boldsymbol{l}^{(1)} = [1,1]$, and $\xi^{(t)} = 0$. There are no noises, and the process is deterministic. The optimal sequence of actions is $i^{(t)} = 1$ for all $t \in [T]$, which incurs the total loss $\frac{1}{2}T(T+1)$. Let $L(t) = l_{i^{(t)}}^{(t)}$ be the loss observed at time $t$. Let $n_i(t)$ be the number of times arm $i$ is chosen at the beginning of time $t$. Let $I_i(t) = \{t_1^{(i)}, t_2^{(i)}, \ldots, t_{n_i(t)}^{(i)}\}$ be the set of times when arm $i$ is chosen up to time $t$. As the loss of the first arm increases by one, $l_1^{(t)} = t$ holds. The loss of the second arm is $l_2^{(t)} = 1 + n_1(t) + 2n_2(t) = t + n_2(t)$. Let $S_i(t) = \sum_{j=1}^{n_i(t)} L(t_j^{(i)})$ be the total loss of arm $i$ at the beginning of time $t$. The empirical mean of the losses of arm $i$ is then given by $\hat{\mu}_i = \frac{S_i(t)}{n_i(t)}$.

$$\hat{\mu}_1(t) = \frac{S_1(t)}{n_1(t)} \tag{76}$$

$$= \frac{1}{n_1(t)} \sum_{j=1}^{n_1(t)} L(t_j^{(1)}) \tag{77}$$

$$\overset{(a)}{=} \frac{1}{n_1(t)} \sum_{j=1}^{n_1(t)} t_j^{(1)} \tag{78}$$

$$= \frac{1}{n_1(t)} \left( \sum_{k=1}^{t-1} k - \sum_{j=1}^{n_2(t)} t_j^{(2)} \right) \tag{79}$$

$$\leq \frac{1}{n_1(t)} \left( \sum_{k=1}^{t-1} k - \sum_{k=1}^{n_2(t)} k \right) \tag{80}$$

$$= \frac{1}{n_1(t)} \left( \frac{t(t-1)}{2} - \frac{n_2(t)(n_2(t)+1)}{2} \right) \tag{81}$$

$$= \frac{1}{n_1(t)} \left( \frac{(n_2(t)+n_1(t)+1)(n_2(t)+n_1(t))}{2} - \frac{n_2(t)(n_2(t)+1)}{2} \right) \tag{82}$$

$$= \frac{1}{2n_1(t)} \left( n_2(t)^2 + 2n_2(t)n_1(t) + n_1(t)^2 + n_2(t) + n_1(t) - n_2(t)^2 - n_2(t) \right) \tag{83}$$

$$= \frac{1}{2n_1(t)} \left( 2n_2(t)n_1(t) + n_1(t)^2 + n_1(t) \right) \tag{84}$$

$$= \frac{1}{2} \left( 2n_2(t) + n_1(t) + 1 \right) \tag{85}$$

$$= \frac{1}{2} \left( t + n_2(t) \right) \tag{86}$$

$$= \frac{t}{2} \left( 1 + \frac{n_2(t)}{t} \right), \tag{87}$$

$$\tag{88}$$

where (a) follows from $l_1^{(t)} = t$, and

$$\hat{\mu}_1(t) = \frac{1}{n_1(t)} \left( \sum_{k=1}^{t-1} k - \sum_{j=1}^{n_2(t)} t_j^{(2)} \right) \tag{89}$$

$$\geq \frac{1}{n_1(t)} \left( \sum_{k=1}^{t-1} k - \sum_{k=1}^{n_2(t)} (t-k) \right) \tag{90}$$

$$= \frac{1}{n_1(t)} \sum_{k=1}^{t-n_2(t)-1} k \tag{91}$$

$$= \frac{1}{n_1(t)} \cdot \frac{(t-n_2(t))(t-n_2(t)-1)}{2} \tag{92}$$

$$= \frac{1}{n_1(t)} \cdot \frac{(t-n_2(t))n_1(t)}{2} \tag{93}$$

$$= \frac{t}{2} \left( 1 - \frac{n_2(t)}{t} \right), \tag{94}$$

$$\tag{95}$$

and thus,

$$\frac{t}{2} \left( 1 - \frac{n_2(t)}{t} \right) \leq \hat{\mu}_1(t) \leq \frac{t}{2} \left( 1 + \frac{n_2(t)}{t} \right). \tag{96}$$

We prove that $t_n^{(2)} \leq 20n \log n$ when $n \geq 19$. $t_{18}^{(2)} = 98$ and $t_{19}^{(2)} = 105 \leq 20 \cdot 19 \log 19 = 1118.88\ldots$. Suppose that $t = t_n^{(2)} \leq 20n \log n$ for some $n \geq 19$. Let $t'$ be the largest index such that $i_{t'} = 1$ and $t' < t$. Since arm 1 is chosen at time $t'$,

$$\hat{\mu}_1(t') \leq \hat{\mu}_2(t') - \sqrt{\frac{2 \log t'}{n_2(t')}} + \sqrt{\frac{2 \log t'}{n_1(t')}} \tag{97}$$

$$\leq \hat{\mu}_2(t') + \sqrt{\frac{2\log t'}{n_1(t')}} \tag{98}$$

$$\overset{(a)}{=} \hat{\mu}_2(t') + \sqrt{\frac{2\log t'}{n_1(t) - 1}} \tag{99}$$

$$\leq \hat{\mu}_2(t') + \sqrt{\frac{2\log t}{n_1(t) - 1}} \tag{100}$$

$$\overset{(b)}{\leq} \hat{\mu}_2(t) + \sqrt{\frac{2\log t}{n_1(t) - 1}}, \tag{101}$$

where (a) follows from the fact that arm 1 is chosen at time $t'$ and not chosen from time $t' + 1$ to $t$ and $n_1(t) = n_1(t') + 1$, and (b) follows from the fact that the losses are monotonic increasing in $t$, and the empirical means are also monotonic increasing in $t$. Since $\hat{\mu}_1(t')$ is not updated from time $t' + 1$ to $t$,

$$\hat{\mu}_1(t) = \hat{\mu}_1(t' + 1) \tag{102}$$

$$= \frac{S_1(t') + L(t')}{n_1(t') + 1} \tag{103}$$

$$= \frac{n_1(t')\hat{\mu}_1(t') + L(t')}{n_1(t') + 1} \tag{104}$$

$$\leq \hat{\mu}_1(t') + \frac{L(t')}{n_1(t') + 1} \tag{105}$$

$$= \hat{\mu}_1(t') + \frac{L(t')}{n_1(t)} \tag{106}$$

$$= \hat{\mu}_1(t') + \frac{t'}{n_1(t)} \tag{107}$$

$$\leq \hat{\mu}_1(t') + \frac{t}{n_1(t)} \tag{108}$$

$$\leq \hat{\mu}_1(t') + \frac{t}{n_1(t) - 1} \tag{109}$$

$$\leq \hat{\mu}_2(t) + \sqrt{\frac{2\log t}{n_1(t) - 1}} + \frac{t}{n_1(t) - 1} \tag{110}$$

$$\leq \hat{\mu}_2(t) + \sqrt{\frac{t}{n_1(t) - 1}} + \frac{t}{n_1(t) - 1} \tag{111}$$

$$= \hat{\mu}_2(t) + \sqrt{\frac{t}{t - n_2(t) - 2}} + \frac{t}{t - n_2(t) - 2} \tag{112}$$

$$\hat{\mu}_2(t + 1) = \frac{S_2(t) + L(t)}{n_2(t) + 1} \tag{113}$$

$$= \frac{n_2(t)\hat{\mu}_2(t) + L(t)}{n_2(t) + 1} \tag{114}$$

$$= \frac{(n_2(t) + 1)\hat{\mu}_2(t) + L(t) - \hat{\mu}_2(t)}{n_2(t) + 1} \tag{115}$$

$$= \hat{\mu}_2(t) + \frac{L(t) - \hat{\mu}_2(t)}{n_2(t) + 1} \tag{116}$$

$$\overset{(a)}{\leq} \hat{\mu}_2(t) + \frac{L(t) - \hat{\mu}_2(t)}{n_2(t)} \tag{117}$$

$$= \hat{\mu}_2(t) + \frac{t + n_2(t) - \hat{\mu}_2(t)}{n_2(t)} \tag{118}$$

$$\overset{(b)}{\leq} \hat{\mu}_2(t) + \frac{t + n_2(t) - \hat{\mu}_1(t) + \sqrt{\frac{t}{t-n_2(t)-2}} + \frac{t}{t-n_2(t)-2}}{n_2(t)} \tag{119}$$

$$\leq \hat{\mu}_2(t) + \frac{t + n_2(t) - \frac{1}{2}(t - n_2(t)) + \sqrt{\frac{t}{t-n_2(t)-2}} + \frac{t}{t-n_2(t)-2}}{n_2(t)} \tag{120}$$

$$= \hat{\mu}_2(t) + \frac{t}{2n_2(t)} + \frac{3}{2} + \frac{\sqrt{\frac{t}{t-n_2(t)-2}} + \frac{t}{t-n_2(t)-2}}{n_2(t)} \tag{121}$$

$$= \hat{\mu}_2(t) + \frac{t}{2n_2(t)} + \frac{3}{2} + \frac{1}{\sqrt{n_2(t)}}\sqrt{\frac{t}{n_2(t)(t - n_2(t) - 2)}} + \frac{t}{n_2(t)(t - n_2(t) - 2)} \tag{122}$$

$$\leq \hat{\mu}_2(t) + \frac{t}{2n_2(t)} + \frac{3}{2} + \sqrt{\frac{t}{n_2(t)(t - n_2(t) - 2)}} + \frac{t}{n_2(t)(t - n_2(t) - 2)} \tag{123}$$

$$\overset{(c)}{\leq} \hat{\mu}_2(t) + \frac{t}{2n_2(t)} + \frac{3}{2} + \sqrt{\frac{t}{t - 3}} + \frac{t}{t - 3} \tag{124}$$

$$\leq \hat{\mu}_2(t) + \frac{t}{2n_2(t)} + \frac{3}{2} + \frac{5}{4} + \frac{5}{4} \tag{125}$$

$$\leq \hat{\mu}_2(t) + \frac{t}{2n_2(t)} + 4 \tag{126}$$

where (a) follows from the fact that the loss is monotone increasing and $L(t) \geq \hat{\mu}_2(t)$, (b) follows from Eq. 112, and (c) follows from the fact that $\frac{1}{n(t-n-2)}$ $(1 \leq n \leq t - 3)$ is minimized when $n = 1$. Suppose $i^{(t+1)} = i^{(t+2)} = \ldots = i^{(t+K)} = 1$. Then, for $k = 1, \ldots, K$,

$$\hat{\mu}_1(t + k + 1) = \frac{n_1(t + k)\hat{\mu}_1(t + k) + L(t + k)}{n_1(t + k) + 1} \tag{127}$$

$$= \hat{\mu}_1(t + k) + \frac{L(t + k) - \hat{\mu}_1(t + k)}{n_1(t + k) + 1} \tag{128}$$

$$\overset{(a)}{\geq} \hat{\mu}_1(t + k) + \frac{L(t + k) - \hat{\mu}_1(t + k)}{t + k} \tag{129}$$

$$= \hat{\mu}_1(t + k) + \frac{t + k - \hat{\mu}_1(t + k)}{t + k} \tag{130}$$

$$\geq \hat{\mu}_1(t + k) + \frac{t + k - \frac{1}{2}(t + k + n_2(t + k))}{t + k} \tag{131}$$

$$= \hat{\mu}_1(t + k) + \frac{1}{2} - \frac{1}{2} \cdot \frac{n_2(t + k)}{t + k} \tag{132}$$

$$= \hat{\mu}_1(t + k) + \frac{1}{2} - \frac{1}{2} \cdot \frac{n_2(t) + 1}{t + k} \tag{133}$$

$$\geq \hat{\mu}_1(t + k) + \frac{1}{2} - \frac{1}{2} \cdot \frac{n_2(t) + 1}{t} \tag{134}$$

$$\geq \hat{\mu}_1(t) + k \left( \frac{1}{2} - \frac{1}{2} \cdot \frac{n_2(t) + 1}{t} \right), \tag{135}$$

where (a) follows from the fact that the loss is monotone increasing and $L(t+k) \geq \hat{\mu}_1(t+k)$ and $n_1(t+1)+1 \leq t + k$. At time $t + K + 1$, the difference of the LCB scores is

$$\text{LCB}_1(t + K + 1) - \text{LCB}_2(t + K + 1) \tag{136}$$

$$\left( \hat{\mu}_1(t + K + 1) - \sqrt{\frac{2\log(t + K + 1)}{n_1(t + K + 1)}} \right) - \left( \hat{\mu}_2(t + K + 1) - \sqrt{\frac{2\log(t + K + 1)}{n_2(t + K + 1)}} \right) \tag{137}$$

$$\geq \hat{\mu}_1(t+K+1) - \sqrt{\frac{2\log(t+K+1)}{n_1(t+K+1)}} - \hat{\mu}_2(t+K+1) \tag{138}$$

$$= \hat{\mu}_1(t+K+1) - \hat{\mu}_1(t) + \hat{\mu}_1(t) - \sqrt{\frac{2\log(t+K+1)}{n_1(t+K+1)}} - (\hat{\mu}_2(t+K+1) - \hat{\mu}_2(t)) - \hat{\mu}_2(t) \tag{139}$$

$$\geq K\left(\frac{1}{2} - \frac{1}{2} \cdot \frac{n_2(t)+1}{t}\right) + \hat{\mu}_1(t) - \sqrt{\frac{2\log(t+K+1)}{n_1(t+K+1)}} - \left(\frac{t}{2n_2(t)} + 4\right) - \hat{\mu}_2(t) \tag{140}$$

$$\geq K\left(\frac{1}{2} - \frac{1}{2} \cdot \frac{n_2(t)+1}{t}\right) - \sqrt{\frac{2\log(t+K+1)}{n_1(t+K+1)}} - \left(\frac{t}{2n_2(t)} + 4\right) \tag{141}$$

$$+ \hat{\mu}_1(t) - \sqrt{\frac{2\log t}{n_1(t)}} - \hat{\mu}_2(t) + \sqrt{\frac{2\log t}{n_2(t)}} + \sqrt{\frac{2\log t}{n_1(t)}} - \sqrt{\frac{2\log t}{n_2(t)}} \tag{142}$$

$$\geq K\left(\frac{1}{2} - \frac{1}{2} \cdot \frac{n_2(t)+1}{t}\right) - \sqrt{\frac{2\log(t+K+1)}{n_1(t+K+1)}} + \sqrt{\frac{2\log t}{n_1(t)}} - \sqrt{\frac{2\log t}{n_2(t)}} \tag{143}$$

$$= K\left(\frac{1}{2} - \frac{1}{2} \cdot \frac{n_2(t)+1}{t}\right) - \left(\frac{t}{2n_2(t)} + 4\right) + \sqrt{\frac{2\log t}{n_1(t)}} - \sqrt{\frac{2\log(t+K+1)}{n_1(t+K+1)}} - \sqrt{\frac{2\log t}{n_2(t)}} \tag{144}$$

$$= K\left(\frac{1}{2} - \frac{1}{2} \cdot \frac{n_2(t)+1}{t}\right) - \left(\frac{t}{2n_2(t)} + 4\right) + \sqrt{\frac{2\log t}{n_1(t)}} - \sqrt{\frac{2\log(t+K+1)}{n_1(t)+K}} - \sqrt{\frac{2\log t}{n_2(t)}}. \tag{145}$$

Here,

$$\frac{2\log t}{n_1(t)} - \frac{2\log(t+K+1)}{n_1(t)+K} = \frac{2}{n_1(t)(n_1(t)+K)}((n_1(t)+K)\log t - n_1(t)\log(t+K+1)) \tag{146}$$

$$= \frac{2}{n_1(t)+K}\left(\left(1 + \frac{K}{n_1(t)}\right)\log t - \log(t+K+1)\right) \tag{147}$$

$$\geq \frac{2}{n_1(t)+K}\left(\left(1 + \frac{K}{t}\right)\log t - \log(t+K+1)\right) \tag{148}$$

$$\geq \frac{2}{n_1(t)+K}\log\frac{t^{1+\frac{K}{t}}}{t+K+1} \tag{149}$$

$$= \frac{2}{n_1(t)+K}\log\frac{t \cdot \exp\left(\frac{K\log t}{t}\right)}{t+K+1} \tag{150}$$

$$\overset{(a)}{\geq} \frac{2}{n_1(t)+K}\log\frac{t(1+\frac{K\log t}{t})}{t+K+1} \tag{151}$$

$$= \frac{2}{n_1(t)+K}\log\frac{t+K\log t}{t+K+1} \tag{152}$$

$$\overset{(b)}{\geq} \frac{2}{n_1(t)+K}\log\frac{t+2K}{t+K+1} \tag{153}$$

$$\geq \frac{2}{n_1(t)+K}\log 1 \tag{154}$$

$$= 0, \tag{155}$$

where (a) follows from $\exp(x) \geq 1 + x$ and (b) follows from $K \geq 1$ and $\log t \geq 2$. Therefore,

$$\mathrm{LCB}_1(t+K+1) - \mathrm{LCB}_2(t+K+1) \geq K\left(\frac{1}{2} - \frac{1}{2} \cdot \frac{n_2(t)+1}{t}\right) - \left(\frac{t}{2n_2(t)} + 4\right) - \sqrt{\frac{2\log t}{n_2(t)}}. \tag{156}$$

Here, we consider two cases.

**Case 1:** $t = t_{n_2}^{(2)} \leq 10n_2(t)$.

$$\frac{t}{3}\left(\frac{1}{2} - \frac{1}{2} \cdot \frac{n_2(t)+1}{t}\right) - \left(\frac{t}{2n_2(t)} + 4\right) - \sqrt{\frac{2\log t}{n_2(t)}} \tag{157}$$

$$\geq \frac{t}{3}\left(\frac{1}{2} - \frac{1}{2} \cdot \frac{n_2(t)+1}{t}\right) - \left(\frac{10n_2(t)}{2n_2(t)} + 4\right) - \sqrt{\frac{2\log 10n_2(t)}{n_2(t)}} \tag{158}$$

$$= \frac{t}{3}\left(\frac{1}{2} - \frac{1}{2} \cdot \frac{n_2(t)+1}{t}\right) - 9 - \sqrt{\frac{2\log 10 + 2\log n_2(t)}{n_2(t)}} \tag{159}$$

$$\geq \frac{t}{3}\left(\frac{1}{2} - \frac{1}{2} \cdot \frac{n_2(t)+1}{t}\right) - 9 - \sqrt{1+1} \tag{160}$$

$$\geq \frac{t}{3}\left(\frac{1}{2} - \frac{1}{2} \cdot \frac{n_2(t)+1}{t}\right) - 11 \tag{161}$$

$$= \frac{t}{3}\left(\frac{1}{2} - \frac{1}{2} \cdot \frac{t-n_1(t)}{t}\right) - 11 \tag{162}$$

$$\overset{(a)}{\geq} \frac{t}{3}\left(\frac{1}{2} - \frac{1}{2} \cdot \frac{t-86}{t}\right) - 11 \tag{163}$$

$$= \frac{t}{3}\left(\frac{43}{t}\right) - 11 \tag{164}$$

$$= \frac{43}{3} - 11 \tag{165}$$

$$> 0, \tag{166}$$

where (a) follows from $n_1(t) \geq n_1(t_{19}^{(2)}) = 86$. Therefore, the LCB algorithm chooses arm 1 at time at most

$$t_{n_2(t)+1}^{(2)} \leq t + \frac{t}{3} + 1 \leq \frac{40}{3}n_2(t) + 1 \leq 20(n_2(t)+1)\log(n_2(t)+1). \tag{167}$$

**Case 2:** $t = t_{n_2}^{(2)} > 10n_2(t)$.

$$\left(\frac{t}{n_2(t)} + 19\right)\left(\frac{1}{2} - \frac{1}{2} \cdot \frac{n_2(t)+1}{t}\right) - \left(\frac{t}{2n_2(t)} + 4\right) - \sqrt{\frac{2\log t}{n_2(t)}} \tag{168}$$

$$= \frac{t}{2n_2(t)} + \frac{19}{2} - \frac{1}{2} - \frac{1}{n_2} - \frac{19n_2(t)}{2t} - \frac{19}{2t} - \left(\frac{t}{2n_2(t)} + 4\right) - \sqrt{\frac{2\log t}{n_2(t)}} \tag{169}$$

$$\geq \frac{t}{2n_2(t)} + \frac{19}{2} - \frac{1}{2} - \frac{1}{n_2} - \frac{19n_2(t)}{20n_2(T)} - \frac{7}{t} - \left(\frac{t}{2n_2(t)} + 4\right) - \sqrt{\frac{2\log t}{n_2(t)}} \tag{170}$$

$$= \frac{t}{2n_2(t)} + \frac{19}{2} - \frac{1}{2} - \frac{1}{n_2} - \frac{19}{20} - \frac{19}{2t} - \left(\frac{t}{2n_2(t)} + 4\right) - \sqrt{\frac{2\log t}{n_2(t)}} \tag{171}$$

$$\geq \frac{t}{2n_2(t)} + \frac{19}{2} - \frac{1}{2} - \frac{1}{19} - \frac{19}{20} - \frac{19}{2 \cdot 105} - \left(\frac{t}{2n_2(t)} + 4\right) - \sqrt{\frac{2\log t}{n_2(t)}} \tag{172}$$

$$\geq \frac{t}{2n_2(t)} + 7 - \left(\frac{t}{2n_2(t)} + 4\right) - \sqrt{\frac{2\log t}{n_2(t)}} \tag{173}$$

$$= 3 - \sqrt{\frac{2\log t}{n_2(t)}} \tag{174}$$

$$\geq 3 - \sqrt{\frac{2\log(20n_2(t)\log n_2(t))}{n_2(t)}} \tag{175}$$

$$\geq 3 - \sqrt{\frac{2\log 20 + 2\log n_2(t) + 2\log\log n_2(t)}{n_2(t)}} \tag{176}$$

$$\geq 3 - \sqrt{1 + 1 + 1} \tag{177}$$

$$> 0. \tag{178}$$

Therefore, the LCB algorithm chooses arm 1 at time at most

$$t^{(2)}_{n_2(t)+1} \leq t + \frac{t}{n_2(t)} + 19 + 1 \tag{179}$$

$$\leq 20 n_2(t)\log n_2(t) + 20\log n_2(t) + 20 \tag{180}$$

$$\leq 20 n_2(t)\log n_2(t) + 20\log n_2(t) + 20 \tag{181}$$

$$= 20(n_2(t)+1)\log n_2 + 20 \tag{182}$$

$$= 20(n_2(t)+1)\log(n_2+1) - 20(n_2(t)+1)\log(n_2+1) + 20(n_2(t)+1)\log n_2 + 20 \tag{183}$$

$$= 20(n_2(t)+1)\log(n_2+1) - 20(n_2(t)+1)\log\left(1+\frac{1}{n_2}\right) + 20 \tag{184}$$

$$\overset{(a)}{\leq} 20(n_2(t)+1)\log(n_2+1) - 20(n_2(t)+1)\left(\frac{\frac{1}{n_2(t)}}{1+\frac{1}{n_2(t)}}\right) + 20 \tag{185}$$

$$= 20(n_2(t)+1)\log(n_2+1) - 20(n_2(t)+1)\frac{1}{n_2(t)+1} + 20 \tag{186}$$

$$= 20(n_2(t)+1)\log(n_2+1) - 20 + 20 \tag{187}$$

$$= 20(n_2(t)+1)\log(n_2+1), \tag{188}$$

where (a) follows from the fact that $\log(1+x) \geq \frac{x}{1+x}$ $(\forall x \geq 0)$.

Therefore, $t^{(2)}_{n_2(t)+1} \leq 20(n_2(t)+1)\log(n_2+1)$ holds for both cases, and we have proved that $t^{(2)}_n \leq 20n\log n$ when $n \geq 19$. Therefore, $t^{(2)}_n \leq 20n\log n \leq 20n\log t^{(2)}_n$, and thus $n_2(t) \geq \frac{t}{20\log t}$. The loss of the LCB algorithm is at lest

$$\frac{1}{2}n_1(t)^2 \boldsymbol{A}_{11} + n_1(t)n_2(t)\boldsymbol{A}_{12} + \frac{1}{2}n_2(t)^2 \boldsymbol{A}_{22} + n_1(t)\left(\boldsymbol{l}^{(1)} - \frac{1}{2}\boldsymbol{A}_{11}\right) + n_2(t)\left(\boldsymbol{l}^{(2)} - \frac{1}{2}\boldsymbol{A}_{22}\right) \tag{189}$$

$$= \frac{1}{2}n_1(t)^2 + n_1(t)n_2(t) + n_2(t)^2 + \frac{1}{2}n_1(t) \tag{190}$$

$$\geq \frac{1}{2}n_1(t)^2 + n_1(t)n_2(t) + n_2(t)^2 \tag{191}$$

$$\geq \frac{1}{2}\left(t - \frac{t}{20\log t}\right)^2 + \left(t - \frac{t}{20\log t}\right)\cdot\frac{t}{20\log t} + \left(\frac{t}{20\log t}\right)^2 \tag{192}$$

$$= \frac{1}{2}t^2 - \frac{t^2}{20\log t} + \frac{t^2}{800\log^2 t} + \frac{t^2}{20\log t} - \frac{t^2}{400\log^2 t} + \frac{t^2}{400\log^2 t} \tag{193}$$

$$= \frac{1}{2}t^2 + \frac{t^2}{800\log^2 t}, \tag{194}$$

Since the loss of the optimal action is $\frac{1}{2}t(t+1)$, the regret is $\frac{t^2}{800\log^2 t} - \frac{1}{2}t = \Omega\left(\frac{t^2}{\log^2 t}\right)$. $\qquad\square$

