# OpenReview forum: "Influential Bandits: Pulling an Arm May Change the Environment"
_TMLR — Accepted by TMLR_

### Review · Reviewer_hdkq · 2025-05-01

**Summary Of Contributions:**

This paper introduces the influential bandit problem, a novel extension of the classical multi-armed bandit framework where selecting one arm can influence the future losses of other arms. The authors begin by analyzing the theoretical properties of this setting, showing that the standard LCB algorithm suffers from a near-quadratic regret lower bound. They also prove the existence of instances where any algorithm must incur at least $\Omega(T)$ regret.

To address this challenge, the authors propose a simple yet effective algorithm, Influential LCB, and show that it achieves near-optimal regret of $O(KT \log T)$ under mild assumptions. They support this result with regret analysis on synthetic datasets. Furthermore, through model analysis on a real-world dataset, they demonstrate that the influential bandit assumption better captures the observed dynamics than the traditional stationary loss assumption.

**Audience:**

Yes

**Broader Impact Concerns:**

As this work is theoretical and centers on a new algorithm for sequential decision-making, there are no apparent ethical concerns arising directly from its content.

**Claims And Evidence:**

Yes

**Requested Changes:**

Please address the weaknesses mentioned in the **Strengths and Weaknesses** section.

Additionally, in Section 4.1, I would be interested to see the regret analysis results when the number of arms is larger (e.g., K = 10).


Minor Changes:
- In Proposition 2.1, clearly define the vector $l^{(1)}$.
- On page 3, make sure matrix $A$ is consistently written in bold. ($A_{horror, horror}, A_{comedy, horror}$)
- On page 5, above equation (13), define $\mathbb{Z}_{*}^K$.
- On page 6, in equation (22), replace $L_2$ with $\mathcal{L}_2$.
- In equations (39) and (40), $g \rightarrow gᵀ$?
- In equation (61), $(2+ 2 ||l^{(1)}||_\infty + 4) \log T \rightarrow (2K+2|| l^{(1)}|| + 4) \log T$  ?
- On page 10, $L^* \rightarrow \mathcal{L}^*$

**Strengths And Weaknesses:**

**Strengths**

1. To the best of my knowledge, this is the first paper to consider a setting where the loss of an arm is influenced by the prior selections of all arms. The motivation for this problem is clear and intuitive.

2. The authors establish regret lower bounds for this new problem setting.

3. The authors propose Influential LCB, a simple yet effective algorithm for this new problem, and provide a thorough theoretical analysis of its regret.

4. The effectiveness of the proposed algorithm and the influential bandit model is empirically validated through experiments, which support the theoretical regret bounds and highlight the significance of the new framework.

5. The discussion section thoroughly addresses the limitations and assumptions of the model, demonstrating strong awareness of potential weaknesses.


**Weaknesses**

1. Model Assumption: The authors assume a symmetric, positive semi-definite interaction matrix to capture the dynamics of arm losses. While they clearly discuss the limitations of the positive semi-definite assumption in the discussion section, I am curious about the necessity of the *symmetric* condition. For example, watching a movie trailer may reduce the loss of the full movie, but watching the full movie might increase the loss (i.e., reduce interest) in the trailer since the user has already seen the content. I wonder how the model performs without the symmetry constraint—specifically, what happens to the MSE in Section 4.2 when the interaction matrix is not assumed to be symmetric?

2. Loss Update Rule: The expected loss of each arm is updated by simply adding the effect from the selected arm via the interaction matrix. However, it may be more realistic to discount the effects of older interactions and place greater weight on more recent ones. For instance, consider watching in the order: movie 1, movie 1, movie 2, ..., movie 100 versus movie 1, movie 2, ..., movie 100, movie 1. According to the model, the final loss would be the same, which seems counterintuitive. Proposition 2.1 is elegant, but the claim that the total loss does not depend on the order of actions feels unrealistic.

3. Readability of the Proof: While I verified the correctness of the proofs, the paper would benefit from sketches or summaries of key proof ideas to improve readability. For example, $x^*$  minimizes the total loss $\mathcal{L}$, while $p^*$ minimizes the quadratic term $\mathcal{L}_2$. The proof converts $x^*$ to $p^*$ to establish an upper bound, but I wonder whether this introduces a gap in tightness.
Is the use of $p^*$ essential for the regret bound? I would be curious to hear the intuition behind introducing $p^*$, which minimizes $\mathcal{L}_2$, instead of directly reasoning about the true minimizer of $\mathcal{L}$.

4. Reference to Frank-Wolfe Algorithm: Since the analysis draws parallels with the Frank-Wolfe algorithm, especially in the convergence behavior, it would be helpful to include references to relevant works on Frank-Wolfe, particularly those covering its convergence analysis.

---

> ### Author Response · Authors · 2025-05-14
>
> We thank the reviewer for their insightful and constructive feedback. We respond to the main points below:
>
> > 1. Model Assumption: The authors assume a symmetric, positive semi-definite interaction matrix to capture the dynamics of arm losses. While they clearly discuss the limitations of the positive semi-definite assumption in the discussion section, I am curious about the necessity of the symmetric condition. For example, watching a movie trailer may reduce the loss of the full movie, but watching the full movie might increase the loss (i.e., reduce interest) in the trailer since the user has already seen the content. I wonder how the model performs without the symmetry constraint—specifically, what happens to the MSE in Section 4.2 when the interaction matrix is not assumed to be symmetric?
>
> We agree that the asymmetric case is both realistic and intriguing. We ran the same setup as in Section 5.1 but with an asymmetric interaction matrix. The resulting MSE was 0.5781 ± 0.8193. This was better than the stationary loss model, but worse than the symmetric case. While this may seem counterintuitive, one plausible explanation is that the effective number of parameters in the asymmetric setting is approximately twice as large, which likely led to overfitting. Interestingly, for the user with the most feedback, the asymmetric model outperformed the symmetric one, suggesting that more flexible asymmetric modeling can be beneficial when data is abundant. Since incorporating asymmetry poses additional theoretical challenges for the quadratic optimization, we leave this as an important direction for future work.
>
> > 2. Loss Update Rule: The expected loss of each arm is updated by simply adding the effect from the selected arm via the interaction matrix. However, it may be more realistic to discount the effects of older interactions and place greater weight on more recent ones. For instance, consider watching in the order: movie 1, movie 1, movie 2, ..., movie 100 versus movie 1, movie 2, ..., movie 100, movie 1. According to the model, the final loss would be the same, which seems counterintuitive. Proposition 2.1 is elegant, but the claim that the total loss does not depend on the order of actions feels unrealistic.
>
> We agree with this point. Indeed, incorporating temporal discounting would capture more realistic phenomena. The challenge is that modeling discounting effects would introduce significant theoretical complexity. We believe our model serves as a reasonable first step toward capturing interaction effects, striking a balance between expressiveness and theoretical tractability. Extending it to incorporate discounting effects would be a promising direction for future work.
>
> > 3. Readability of the Proof
> > The proof converts $x^*$ to $p^*$ to establish an upper bound, but I wonder whether this introduces a gap in tightness.
>
> The transformation from $x^\ast$ to $p^\ast$ does not introduce a gap in the regret bound: the loss difference between $\mathcal{L}(x^\ast)$ and $\mathcal{L}_2(p^\ast)$ is only $O(T)$, which is dominated by the leading regret term $O(T \log T)$.
>
> >  Is the use of $p^*$ essential for the regret bound?
>
> While $p^\ast$ is not strictly necessary, it is conceptually helpful. The key idea in our analysis is to interpret the trajectory of states as an optimization path. However, $x_1 \in 1 \Delta_d, x_2 \in 2 \Delta_d, \ldots, x_T \in T \Delta_d$ grow, and the domain changes over time and cannot be treated as a single optimization problem. By contrast, all of $p_1, p_2, \ldots, p_T \in \Delta_d$ stay in the probabilistic simplex, allowing us to interpret the process as optimizing $\min_p \mathcal{L}_2(p) \text{ s.t. } p \in \Delta_d$. We initially developed a version of the proof without using $p^\ast$, but found it less intuitive. We therefore chose the current version for clarity.
>
> > 4. Reference to Frank-Wolfe Algorithm: Since the analysis draws parallels with the Frank-Wolfe algorithm, especially in the convergence behavior, it would be helpful to include references to relevant works on Frank-Wolfe, particularly those covering its convergence analysis.
>
> Thank you for the suggestion. We will include references to the Frank-Wolfe algorithm literature in the camera ready.
>
> > Additionally, in Section 4.1, I would be interested to see the regret analysis results when the number of arms is larger (e.g., K = 10).
>
> Thank you for the proposal. We ran the same experiments with K = 10. The slope for the standard LCB was 1.58, and the slope for the Influential LCB was 1.19. Although the results were less sharp than those for K = 3, the same tendency was observed, i.e., the regret of the standard LCB was of the order of near-quadratic, and the regret of the influential LCB was of the order of near-linear.

---

### Review · Reviewer_kjw6 · 2025-05-11

**Summary Of Contributions:**

This work proposes a new nonstationary bandit setting called the influential bandit, where selecting one arm may affect the reward of others, including both positive and negative influence. It models the influence with a positive semi-definite matrix, and therefore, the cumulative loss can be represented as a quadratic function. The paper proposes several theoretical results in this setting.

First, the trivial upper bound is $O(T^2)$, which differs from traditional bandits. Second, the traditional UCB algorithm does not work well, as it will cause $\Omega(T^2/(\log T)^2)$ regret. Third, it constructs a hard instance to show that any early mistake can cause huge influence on the later exploration, incuring a $\Omega(T)$ lower bound. Finally, it proposes a modified LCB algorithm that achieves $O(KT\log T)$ regret.

Experiments are conducted to validate the theoretical observation  on both synthetic
 and real-world datasets.

**Audience:**

Yes

**Broader Impact Concerns:**

This work introduces an interesting setting for handling the influence of past actions in the decision-making under uncertainty problem. It may be interesting to other researchers working on bandits and reinforcement learning.

**Claims And Evidence:**

Yes

**Requested Changes:**

I think the writing is pretty good, and nothing needs to be changed. But I have some questions on this topic.

1. I think the current solution does not consider the effect of noise. In the classical UCB algorithm, the UCB term $\sqrt{\log t/n_i}$ is incorporated to address the randomness of noise. However, in the influential LCB algorithm, the estimated loss is directly chosen to be that at the current step, without using the information from previous rounds, and no similar $\sqrt{\log t/n_i}$ term due to the noise. Is my understanding correct?

2. Other than UCB, another classical algorithm for bandits is Thompson sampling (TS). How do you think TS will behave in this setting?

3. In classical bandits, a lower bound dependent on the number of actions $K$ can be proved. Is it also possible in the influential bandit setting?

I think it would be better if these questions could be further discussed. But if not, the paper is still very good.

**Strengths And Weaknesses:**

Strengths:

1. It proposes an interesting bandit setting, which models the interactive influence between actions.

2. It provides a complete theoretical analysis of this problem. The writing is very clear, and all the confusion caused by its difference from the traditional setting has been covered in the presentation.

3. It proposes a new algorithm, which can achieve nontrivial regret guarantees on this problem, and it outperforms the classical UCB algorithm.

4. The experiments are very convincing, validating the correctness of the theory.

I do not find any obvious weakness

---

> ### Author Response · Authors · 2025-05-14
>
> We thank the reviewer for their encouraging feedback. Below, we respond to each of the points raised:
>
> > I think the current solution does not consider the effect of noise. In the classical UCB algorithm, the UCB term is incorporated to address the randomness of noise. However, in the influential LCB algorithm, the estimated loss is directly chosen to be that at the current step, without using the information from previous rounds, and no similar term due to the noise. Is my understanding correct?
>
> Yes, your understanding is correct. The current version of the algorithm does not explicitly incorporate noise terms, unlike standard UCB algorithms. This is because the interaction effects dominate the regret in our setting, and noise contributes only a linear or sublinear term. Our proposed algorithm focuses on addressing this dominant factor. That said, as we discuss in Section 5.2, accounting for noise may improve finite-time performance, and as discussed in Section 5.3, noise-aware strategies may be necessary to achieve sublinear regret in some instances. Thus, extending the algorithm to more explicitly model noise is a promising direction for future work.
>
> > Other than UCB, another classical algorithm for bandits is Thompson sampling (TS). How do you think TS will behave in this setting?
>
> We expect that Thompson Sampling, like standard LCB, would suffer from near-quadratic regret in our setting. This is because traditional TS, like standard LCB, only updates the posterior of the pulled arm (i.e., the arm that emits a loss observation), without accounting for how pulling that arm affects the losses of other arms. This is in contrast to our proposed algorithm, which updates all arms' states. Thus, we believe TS would face the same drawback as standard LCB in the influential bandit setting.
>
> > In classical bandits, a lower bound dependent on the number of actions can be proved. Is it also possible in the influential bandit setting?
>
> Thank you for raising this important point. Establishing a lower bound on the number of actions is indeed a meaningful and challenging direction. At present, we have not succeeded in deriving such a bound, and we consider this an important open question for future research.

---

### Review · Reviewer_S4bi · 2025-05-11

**Summary Of Contributions:**

This paper introduces a new problem of influential bandits where pulling one arm may affect the rewards of all arms in the future. To formulate this problem, this work introduces a correlation matrix and a new algorithm based on the LCB estimator tailored to the structure of the dynamic loss. This paper also gives the lower bound under this problem setting and showcases the improved regret of their proposed LCB algorithm. Some experimental results are finally presented.

**Audience:**

Yes

**Broader Impact Concerns:**

No ethical implications and this work is mostly theoretical.

**Claims And Evidence:**

Yes

**Requested Changes:**

Please refer to the weakness section above.

**Strengths And Weaknesses:**

Strengths:
1. The presentation of this work is mostly clear and easy to understand.
2. This work introduces an interesting problem when pulling one arm will affect the rewarding structure of all the other arms in the long run.
3. Both upper bound and lower bound are presented.
4. The experimental results are also included to highlight the efficiency of the proposed LCB method.

Weaknesses:
1. For presentation, some parts can be improved: 1. This paper uses UCB and LCB interchangeably, such as in the abstract. While I can understand that you are trying to mention the same thing when the response is reward/loss, it still feels a little bit confusing. 2. Some expression should be revised, e.g. "not uncommon" to "common" in Section 1.1.
2. For the problem setting, especially the lower bound of the regret is in the order of T, which seems surprising. I understand your proof of Proposition 2.3. But I feel it is a little bit weird that pulling the wrong arm one time will lead to the linear regret bound in the end. Do you think it is more practical to set this correlation effect only over the next few nearby rounds?
3. In your problem setting, the total loss does not depend on the order of actions. Can you justify this key assumption? Under that condition, repeating the horror movies for the first ten rounds and then comedies in the next ten rounds has the same effect for the further recommendation as recommending horror and comedy movie in term over the total twenty rounds. However, I think they should be drastically different in applications. I feel the correlation effect should gradually decay over time, which would be more reasonable. This is also why your lower bound is T rather than sqrt(T) as the correlation impact is currently too strong.
4. It is better to highlight the difference between your Algorithm 1 and the traditional LCB with a couple of sentences and illustrate why it works over the traditional LCB.

---

> ### Author Response · Authors · 2025-05-14
>
> We thank the reviewer for their thoughtful comments and helpful suggestions. Below, we address each point raised.
>
> > For presentation, some parts can be improved: 1. This paper uses UCB and LCB interchangeably, such as in the abstract. While I can understand that you are trying to mention the same thing when the response is reward/loss, it still feels a little bit confusing. 2. Some expression should be revised, e.g. "not uncommon" to "common" in Section 1.1.
>
> Thank you for pointing them out. We will fix them in the camera ready.
>
> > For the problem setting, especially the lower bound of the regret is in the order of T, which seems surprising. I understand your proof of Proposition 2.3. But I feel it is a little bit weird that pulling the wrong arm one time will lead to the linear regret bound in the end. Do you think it is more practical to set this correlation effect only over the next few nearby rounds?
>
> We believe our setting is reasonable for the following reasons:
>
> (i) In real-world applications, some decisions have long-term, irreversible consequences. For instance, taking an offensive action early may cause a long-lasting negative impression on that person or service, resulting in regret that scales with time.
>
> (ii) It is important to note that not all instances of the influential bandits lead to linear regret from a single wrong action. Proposition 2.3 states that there *exists* such an instance. If the interaction matrix $A$ has an eigenvalue 0 with a corresponding non-negative eigenvector $v$, then cycling between actions proportional to $v$ can neutralize the accumulated interaction effects, resulting in bounded-time interaction effects. In such instances, actions are recoverable. As observed in Section 5.3, there exists instances where the regret scales sublinearly as well as instances where the regret scales linearly.
>
> Therefore, the influential bandit framework can capture both irreversible and recoverable influence dynamics, which we believe is one of its strengths. That said, we agree that modeling influence with temporal decay is another reasonable formulation. We view both settings as valuable, and each could merit a dedicated study.
>
> > In your problem setting, the total loss does not depend on the order of actions. Can you justify this key assumption? Under that condition, repeating the horror movies for the first ten rounds and then comedies in the next ten rounds has the same effect for the further recommendation as recommending horror and comedy movie in term over the total twenty rounds. However, I think they should be drastically different in applications. I feel the correlation effect should gradually decay over time, which would be more reasonable. This is also why your lower bound is T rather than sqrt(T) as the correlation impact is currently too strong.
>
> This concern is closely related to the previous one. We agree that incorporating temporal decay into the influence model would yield more realistic behavior in some applications. However, this would greatly complicate the theoretical analysis. We consider our current model a reasonable first step that balances analytical tractability and the ability to capture interactions. Developing models with decaying influence is an important direction for future work.
>
> > It is better to highlight the difference between your Algorithm 1 and the traditional LCB with a couple of sentences and illustrate why it works over the traditional LCB.
>
> Thank you for the suggestion. The key distinction lies in the update rule: Standard LCB updates the state of only one arm, i.e., the pulled arm, at a time, whereas Algorithm 1 updates the states of all arms, taking into account the interaction effects.

---

> > ### Comment · Reviewer_S4bi · 2025-05-14
> > **Thanks for your response**
> >
> > Thank you for your quick response. I have another concern regarding the difference between the standard LCB and your proposed one: in practice since we are not aware of whether the correlation exists between different arms, then how should we choose when to use the standard one and when to use the proposed one? Can your method attain the similar practical performance as the standard LCB when there is no correlation between arms and the rewards at each iteration are iid generated?

---

> > > ### Author Response · Authors · 2025-05-14
> > >
> > > Thank you for your follow-up question. There are two main approaches to deciding whether the influential method should be used in practice:
> > >
> > > 1. Data-driven estimation of interactions: As shown in Section 4.2, the interaction matrix $A$ can be estimated from observed data. If the estimated $A$ is sufficiently large, using influential methods is recommended.
> > >
> > > 2. Meta-level selection: In practice, one can carry out an A/B test. Alternatively, one can treat the choice between the standard and influential methods as a meta-decision and apply an online learning algorithm to choose between the two as meta-action.
> > >
> > > Unfortunately, our proposed method does not perform as well as standard LCB when there are no interaction effects and the losses are i.i.d. In this sense, the situation is analogous to how LCB performs worse in adversarial environments while EXP3 underperforms in stochastic environments. Developing a unified algorithm that performs well both when interactions are present and when they are absent to establish a best-of-both-worlds scenario is an important direction for future research.

---

### Decision · Action_Editor_jaNf · 2025-06-11

**Recommendation:** Accept with minor revision

**Additional Comments:**

The paper presents a novel and well-motivated setting with sound theoretical analysis and empirical support. However, minor revisions are needed to clarify the practicality of the assumptions and to better distinguish the contribution from related work. A brief discussion of when the model is applicable and possible extensions would strengthen the paper.

**Audience:**

Yes

**Audience Explanation:**

Researchers in TMLR’s audience who focus on online learning, reinforcement learning, non-stationary environments, or theoretical analysis of bandits would likely find the paper’s problem formulation, theoretical bounds, and algorithm design valuable and relevant.

**Claims And Evidence:**

Yes

**Claims Explanation:**

All reviewers agree that the claims are backed by correct and well-presented theoretical analysis and experiments, satisfying the expectation of clarity and support.